# Staggered circular nanoporous graphene converts electromagnetic waves into electricity

Hualiang Lv[1], Yuxing Yao[2], Shucong Li[3], Guanglei Wu[4,5], Biao Zhao[4,6], Xiaodi Zhou[1], Robert L. Dupont[1], Ufuoma I. Kara[1], Yimin Zhou[7], Shibo Xi[8], Bo Liu[9] ✉, Renchao Che[4,6,10] ✉, Jincang Zhang[10], Hongbin Xu[11], Solomon Adera[7], Renbing Wu[4] ✉ & Xiaoguang Wang[1,12] ✉

Harvesting largely ignored and wasted electromagnetic (EM) energy released by electronic devices and converting it into direct current (DC) electricity is an attractive strategy not only to reduce EM pollution but also address the ever-increasing energy crisis. Here we report the synthesis of nanoparticle-templated graphene with monodisperse and staggered circular nanopores enabling an EM–heat–DC conversion pathway. We experimentally and theoretically demonstrate that this staggered nanoporous structure alters graphene's electronic and phononic properties by synergistically manipulating its intralayer nanostructures and interlayer interactions. The staggered circular nanoporous graphene exhibits an anomalous combination of properties, which lead to an efficient absorption and conversion of EM waves into heat and in turn an output of DC electricity through the thermoelectric effect. Overall, our results advance the fundamental understanding of the structure–property relationships of ordered nanoporous graphene, providing an effective strategy to reduce EM pollution and generate electric energy.

The recent breakthroughs in information and communication technologies (e.g., 3G–5G) have led to an astronomical rise in electronic device usage and the release of particularly low-frequency electromagnetic (EM) waves, mainly ranging from 2 to 5 GHz (L- and C-band), into the surrounding environment[1,2]. However, only 20–30% of EM waves (e.g., produced by base stations, wireless routers and electronic devices) are utilized in telecommunications[3,4]. The rest are largely ignored and wasted, causing high levels of EM pollution in the surrounding environment and leading to health concerns, signal interference in device-to-device communications, and heat generation in electronic devices (known as overheating)[5,6]. Therefore, harvesting the freely available EM radiation and converting it into usable direct current (DC) electricity is an attractive strategy to not only reduce EM pollution but also address the ever-increasing energy crisis. To date, no single-component material system can convert EM waves to DC electricity because of contradictory material requirements[7-9]. For example,

[1]Willian G. Lowrie Department of Chemical and Biomolecular Engineering, The Ohio State University, Columbus, OH 43210, USA. [2]Division of Chemistry and Chemical Engineering, California Institute of Technology, Pasadena, CA 91125, USA. [3]Department of Chemistry and Chemical Biology, Harvard University, Cambridge, MA 02138, USA. [4]Department of Materials Science, Fudan University, Shanghai 200433, P. R. China. [5]College of Materials Science and Engineering, Qingdao University, Qingdao 266071, P. R. China. [6]Laboratory of Advanced Materials, Shanghai Key Lab of Molecular Catalysis and Innovative Materials, Academy for Engineering & Technology, Fudan University, Shanghai 200438, P. R. China. [7]Department of Mechanical Engineering, University of Michigan, Ann Arbor, MI 48109, USA. [8]Institute of Chemical and Engineering Sciences, A*STAR, 627833 Singapore, Singapore. [9]College of Mechanical and Vehicle Engineering, Hunan University, Changsha 410082, P. R. China. [10]Zhejiang Laboratory, Hangzhou 311100, P. R. China. [11]Department of Materials Science and Engineering, Massachusetts Institute of Technology, Cambridge, MA 02139, USA. [12]Sustainability Institute, The Ohio State University, Columbus, OH 43210, USA. ✉e-mail: boliu@hnu.edu.cn; rcche@fudan.edu.cn; rbwu@fudan.edu.cn; wang.12206@osu.edu

EM absorbing and shielding materials require a high permittivity, which is typically associated with a high thermal conductivity, while thermoelectric materials require a low thermal conductivity and fail to absorb EM radiation[10,11]. To overcome this challenge, we sought to design a single-component material system that enables EM–heat–DC conversion: (i) efficiently absorbing and converting low-frequency EM waves into heat, creating an associated temperature gradient, and (ii) outputting DC electricity via the Seebeck effect.

Graphene, with its characteristic high permittivity, has been demonstrated to possess remarkable EM dissipation abilities (e.g., converting EM into heat)[12,13]. However, the intrinsically high thermal conductivity, low Seebeck coefficient, and zero bandgap structure prevent heat–DC conversion[14,15]. Element doping and the creation of graphene nanostructures (e.g., nanoribbons) have been widely used to alter graphene's electronic and phononic structure by covalently tuning the intralayer atomic bonding[16–18]. While not fully understood yet, recent research has shown that the manipulation of the interlayer interactions between different graphene layers (e.g., van der Waals forces) by stacking two sheets of graphene that are twisted by a small angle (known as the magic angle of graphene superlattices) enables a variety of material properties and functions[19,20]. In order to achieve our goal, we sought to tune the electronic and phononic structure of graphene via a combination of both intralayer and interlayer strategies – specifically, the creation of ordered nanopores in graphene surfaces (intralayer effect) and the formation of partially overlapped nanopores on different graphene layers (namely a staggered porous structure; the interlayer effect is similar to the effect from magic angle graphene superlattices). Current methods for creating ordered porous graphene, such as electron beams, can only achieve micrometer-sized, completely overlapped pores, which usually display a limited density of carbon atoms located at the pore edges and thus a weak intralayer effect[21–23]. Therefore, the synthesis of monodisperse, nanometer-sized pores (<10 nm) with a staggered porous structure across different graphene layers remains challenging.

To overcome this challenge, this work reports a method to create monodisperse, nanometer-sized circular pores with well-controlled pore sizes and shapes on graphene templated by transition metal oxide nanoparticles formed in-situ. In graphene with more than one layer, the nanopores on different graphene layers partially overlap with each other, resulting in a desirable staggered nanoporous

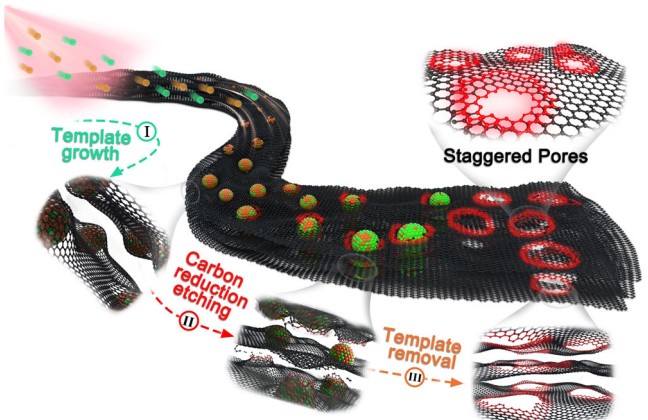

**Fig. 1 | Schematic illustration of the synthesis of graphene with ordered circular nanopores.** **I** In-situ growth of monodisperse spherical $Fe_3O_4$ nanoparticles on a graphene surface (green and orange spheres represent iron and oxygen atoms, respectively). **II** In-situ etching of graphene templated by $Fe_3O_4$ nanoparticles. **III** Formation of nanometer-sized circular pores in graphene after removing the $Fe_3O_4$ nanoparticles (red highlights are carbon atoms at the edge of the nanopores). We note here that the nanopores in different graphene layers are staggered in graphene with more than one layer.

structure. We found that the formed non-graphitized carbon at the edge of the graphene nanopores serves as dipoles to improve the EM–heat conversion through dipole polarization relaxation at relatively low EM frequencies (i.e., 2–5 GHz). Furthermore, the pore edges promote phonon scattering to reduce the thermal conductivity of the graphene and confine the electron transport by splitting the Dirac point and breaking up the Fermi energy surfaces, which significantly enhances the Seebeck effect of graphene. As a result, the synergy of the high permittivity, the reduced thermal conductivity and the enhanced Seebeck coefficient makes this class of staggered, ordered nanoporous graphene a promising material to achieve the proposed EM–heat–DC conversion.

## Results

### Synthesis and characterization of nanoporous graphene

To achieve the desired electronic structure for DC generation from EM waves, we sought to create monodisperse, nanometer-sized circular pores in graphene surfaces using spherical nanoparticles as templates, as shown in Fig. 1. First, monodisperse, spherical iron (II, III) oxide ($Fe_3O_4$) nanoparticles were grown in-situ on the surface of the graphene nanosheets via a thermal decomposition process. $Fe_3O_4$ nanoparticles oxidized the underlying graphene during the subsequent annealing treatment to locally generate nanopores. Finally, the remaining $Fe_3O_4$ nanoparticles were removed using hydrochloric acid, revealing these monodisperse nanometer-sized pores in the graphene surfaces as shown in the representative transmission electron microscopy (TEM) images in Fig. 2a, b and Supplementary Fig. 1. The phase evolution of the oxidation process was verified using X-ray photoelectron spectroscopy (XPS) and X-ray diffraction (XRD), as shown in Supplementary Fig. 2. In addition, square and hexagonal nanopores can be created on graphene using cubic and hexagonal $Fe_3O_4$ nanoparticles, respectively (Fig. 2c and Supplementary Figs. 3–4).

Besides tuning the shape and size of the nanopores in the graphene surface, we have also created nanopores in multilayer structures of graphene. Representative TEM images show bilayer, trilayer, and multilayer (six layers on average) graphene with circular nanopores (Fig. 2d–f). In particular, the $Fe_3O_4$ nanoparticles initially grown on the outer layers of graphene successively etched carbon atoms from the subsequent layers, which is combined with particle-induced local distortions and the interlayer slipping of graphene to cause a partial overlapping of nanopores on different graphene layers (namely staggered pore structures), as evidenced in the inset in Fig. 2d. These experimental results demonstrate the ability to generate staggered homogeneous nanoporous graphene with controllable pore sizes, shapes, and layer numbers.

### Electromagnetic wave dissipation by nanoporous graphene

For efficient DC electricity generation from EM waves released from 3G–5G electronics, efficient absorption and conversion of EM energy into thermal energy at low EM frequencies (typically 2–5 GHz; Supplementary Table 1) is required, which is quantified by the EM dissipation factor ($\eta$). The heat ($W$) generated by the EM dissipation can be calculated as[24]:

$$W = D \cdot E^2 \cdot f \cdot \eta \tag{1}$$

where $D$ is a coefficient associated with the volume of EM dissipation materials, and $E$ and $f$ represent the power and frequency of the applied EM wave, respectively. $\eta$ can be written as[25]:

$$\eta = \varepsilon_r \cdot \tan\delta_e = \varepsilon_r \cdot \varepsilon''/\varepsilon' \tag{2}$$

where $\varepsilon_r$ is the relative complex permittivity and $\tan\delta_e$ is the dielectric loss tangent, defined as the ratio of the imaginary and real part of the

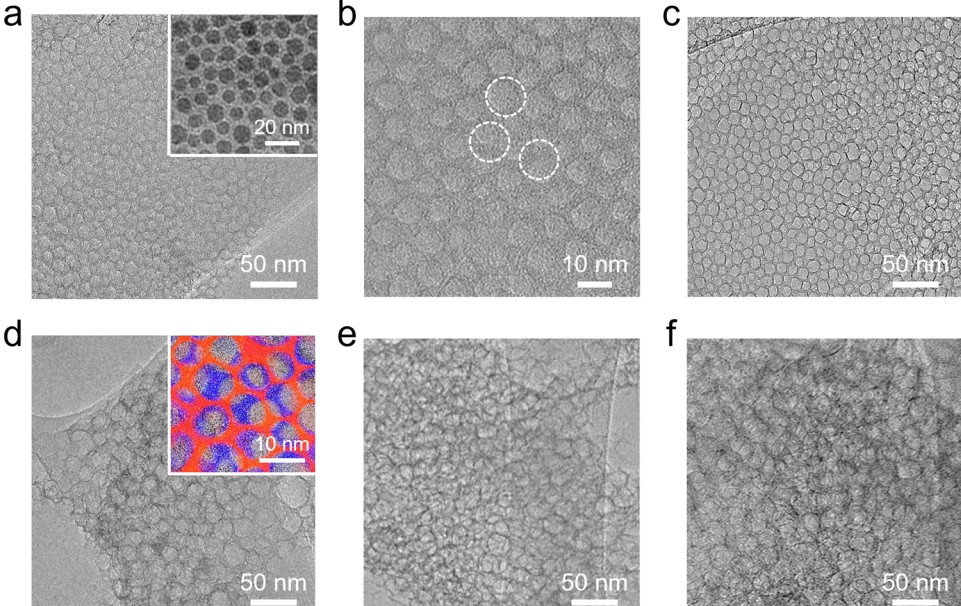

**Fig. 2 | Morphology of graphene with ordered nanometer-sized pores.** Representative TEM images of (**a**, **b**) monolayer graphene with circular pores with a diameter of $6 \pm 1$ nm, (**c**) monolayer graphene with square nanopores with an edge length of $7 \pm 1$ nm, (**d**) bilayer, (**e**) trilayer, and (**f**) multilayer graphene with circular nanopores. Inset in (**a**) shows a representative TEM image of the spherical $Fe_3O_4$ nanoparticles. The staggered nanoporous structure was observed on graphene with more than one layer. The top and bottom layers in the bilayer nanoporous graphene in the inset in (**d**) were highlighted in red and blue, respectively.

permittivity ($\varepsilon''/\varepsilon'$). The permittivity and $\eta$ of graphene with different structures are shown in Fig. 3a and Supplementary Figs. 5–6. Pristine graphene fabricated by chemical vapor deposition (CVD) possesses a limited number of defects, and thus lacks sufficient dipoles largely restricting the relaxation behavior in response to GHz-frequency EM fields. Within the EM frequency range of 2–5 GHz, the $\eta$ of monolayer and bilayer nanoporous graphene is larger than their trilayer and multilayer counterparts. In addition, the $\eta$ of graphene with circular nanopores is higher than that of pristine graphene, which is attributed to the excellent EM wave incident angle-independence of the circular pores. We note here that the highest $\eta$ values among the ordered nanoporous graphene, 160.8 for bilayer graphene with staggered circular pores, is higher than that of nonporous graphene (116.9) and a majority of the conventional EM dissipating materials (Supplementary Table 2).

Our measurement of the Cole–Cole curves of ordered nanoporous graphene (Supplementary Fig. 7) demonstrates that the remarkable $\eta$ of ordered nanoporous graphene is significantly attributed to the dipole polarization relaxation at low EM frequencies, which is consistent with the Debye relaxation theory[26,27]. Considering the fact that the dipole polarization relaxation of conventional EM dissipating materials typically takes place at high frequencies (usually > 8 GHz) due to a comparable relaxation time of their dipoles with respect to the frequency of the applied EM field (see Supplementary Table 2), we performed XPS, Raman spectroscopy, and Fourier transform infrared spectroscopy (FT-IR) analysis of the ordered nanoporous graphene, which collectively demonstrated that the distribution of dipoles with large electronegativity differences (e.g., C–O, C=O and C–OH) at the pore edges increases the relaxation time of the dipoles and thus shifts the dipole polarization relaxation toward lower frequencies[28,29], as shown in Supplementary Fig. 8 and Supplementary Note 1. The above proposed mechanism is further supported by the low $\eta$ of graphene at low EM frequencies with polydisperse, distorted pores, nitrogen or sulfur-doped graphene, and graphene with completely overlapped nanopores across different graphene layers, as the dipole polarization relaxations are located at medium and high EM frequencies (> 5 GHz) and not at low frequencies (2–5 GHz), as shown in Supplementary

Fig. 9 and Supplementary Notes 2–3. Furthermore, we measured an increase in the intensity of the dipole polarization relaxation with temperature, leading to an increase in $\eta$ of ordered porous graphene (Supplementary Fig. 10). This result suggests that the ordered nanoporous graphene enables efficient EM dissipation at elevated temperatures.

## Electrical and thermal conductivity of nanoporous graphene

The presence of a high density of dipoles at the pore edges in the graphene lowers the number of graphitized carbon atoms and affects the transport of electrons. As shown in Fig. 3b and Supplementary Fig. 11a–h, the mobility and density of Hall carriers in ordered nanoporous graphene are lower than in nonporous bilayer graphene. As a result, the electrical conductivity ($\sigma$) of ordered nanoporous graphene lies between 3000 and 1000 S/cm over a temperature range of 300–500 K, which is ~20–30% that of nonporous bilayer graphene, as shown in Fig. 3c and Supplementary Fig. 11i–l.

Upon dissipating the EM waves into heat, an ultralow thermal conductivity ($\kappa_T$) is desired to maintain the temperature profile in the ordered nanoporous graphene to drive the directional diffusion of charge carriers from the hot side to the cold side. However, pristine (nonporous) graphene is well known for its ultrahigh $\kappa_T$, preventing the formation of a sufficient temperature gradient for the generation of DC electricity via the Seebeck effect[30]. In the next set of experiments, we sought to investigate the effect of the nanoporous structure on $\kappa_T$. As evidenced in Fig. 3d, the $\kappa_T$ of bilayer graphene with circular nanopores lies between 3.5 and 2.1 W/m·K over a temperature range of 300–500 K, which is two orders of magnitude lower than that of nonporous bilayer graphene. We also observed that $\kappa_T$ decreases with an increase in the number of graphene layers. We note here that the $\kappa_T$ of bilayer graphene with circular nanopores is almost two orders of magnitude lower than graphene nanoribbons, other two-dimensional materials (e.g., tungsten diselenide, 2H molybdenum disulfide, and MXenes), and porous nanomaterials (e.g., graphene aerogel, graphitized carbon, and carbon foam), as listed in Supplementary Table 3. The $\kappa_T$ of graphene with different layers is also compared in Supplementary Fig. 12.

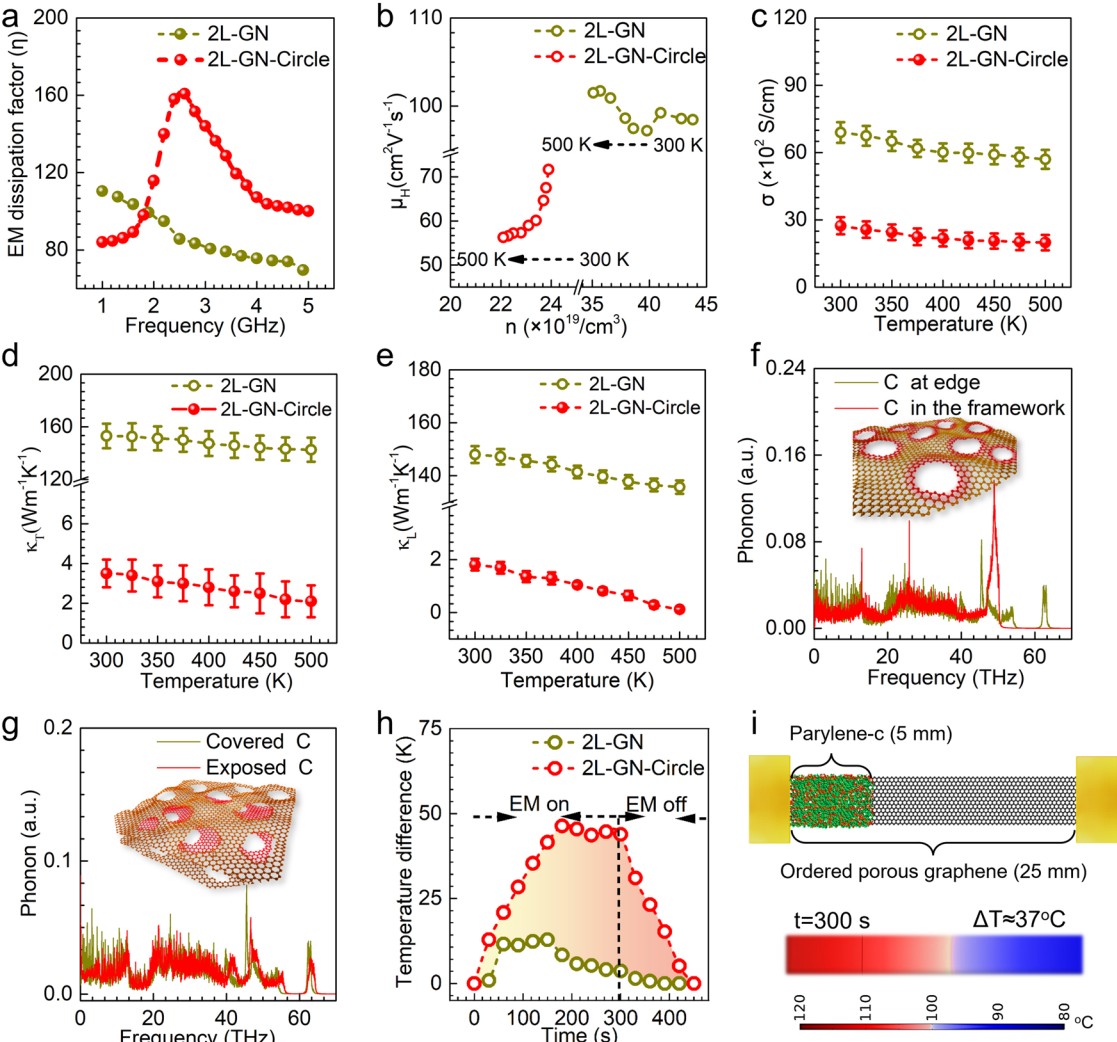

**Fig. 3 | Effect of staggered, ordered circular nanopores on the electromagnetic (EM) dissipation, electrical conductivity, and thermal conductivity of graphene. a** EM frequency-dependent EM dissipation factor ($\eta$) of bilayer graphene with (2L-GN-Circle) and without (2L-GN) ordered circular nanopores. **b** Mobility ($\mu_H$) of bilayer graphene with and without ordered circular nanopores as a function of Hall carrier density ($n$). Arrows in (**b**) correspond to a temperature increase from 300 K to 500 K. **c** Electric conductivity ($\sigma$), (**d**) thermal conductivity ($\kappa_T$), and (**e**) lattice thermal conductivity ($\kappa_L$) of bilayer graphene with and without ordered circular nanopores as a function of temperature. Error bars represent standard deviations from three independent measurements. Molecular dynamics simulation of the phonon density of states of (**f**) monolayer and (**g**) bilayer graphene with ordered circular nanopores. Inset images in (**f**) and (**g**) show the schematic of monolayer and bilayer porous graphene structures, respectively. In (**f**), the yellow and red lines represent the carbon atoms located at the pore edges and in the framework of monolayer nanoporous graphene, respectively. In (**g**), the yellow and red lines represent the exposed carbon atoms and the carbon atoms covered by neighboring layers of graphene, respectively. **h** Temperature difference ($\Delta T$) of graphene with ordered circular nanopores as a function of time. EM radiation is applied from 0 to 300 s. **i** Finite-element simulation of the $\Delta T$ of graphene with ordered circular nanopores during EM radiation. The staggered, ordered nanopores induce a dipole polarization relaxation at lower EM frequencies and increase the $\eta$ of graphene, weakening the phonon coupling behavior and decreasing the $\kappa_T$ of graphene.

It is well established that $\kappa_T$ has contributions from the electron and lattice thermal conductivity ($\kappa_e$ and $\kappa_L$, respectively) and that $\kappa_e$ is proportional to $\sigma$[31]. To elucidate the role of the nanoporous structure in the reduced $\kappa_T$, we measured the $\kappa_e$, and $\kappa_L$ of the ordered nanoporous graphene as a function of temperature (see Supplementary Note 4). As shown in Fig. 4e and Supplementary Fig. 13, upon creating the nanopores on graphene, an abrupt decrease in $\kappa_L$ was found to be the main contributor to the reduction of $\kappa_T$. To provide insight, we performed molecular dynamics simulations to compute the localization and scattering of phonons at the pore edges of graphene (see interatomic potential parameters and detailed computational methods in Supplementary Note 5, Supplementary Fig. 14–19 and Supplementary Table 4). In one graphene layer, carbon atoms in the dipoles located at the pore edges weaken the phonon coupling behavior, particularly in the phonon frequency of 50–65 THz, as evidenced in Fig. 3f and Supplementary Figs. 15–16. The mismatch in the phonon density of states at a phonon frequency <50 THz also gives rise to strong phonon scattering at the pore edges, known as the phonon backscattering effect[32,33]. In addition, the existence of carbon atoms both covered by neighboring layers of graphene and exposed by the pores weakens the phonon coupling behavior in ordered nanoporous graphene with more than one layer, as shown in Fig. 3g (see Supplementary Figs. 17–20 for the effect of the number of graphene layers and staggered porous graphene with different overlap ratios, which are defined by the ratio of the pore area covered by a neighboring graphene layer to the total pore area). The presence of nanopores was found to decrease the spacing between graphene layers (evidenced by the diffraction peak of the (002) crystal plane of graphene in Supplementary Fig. 20), further weakening the phonon coupling of covered and exposed carbon atoms in a single graphene layer.

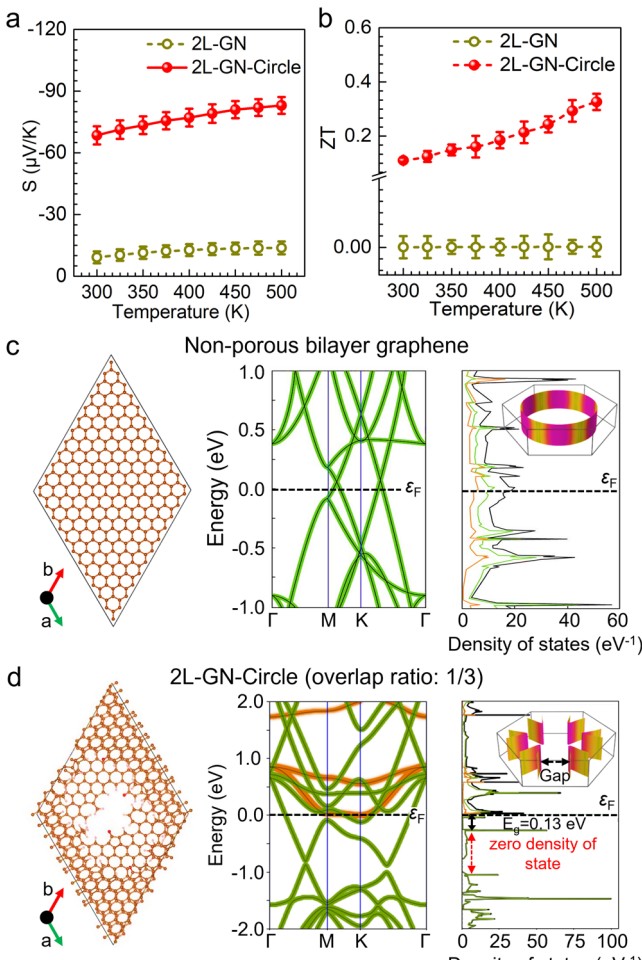

**Fig. 4 | Effect of staggered, ordered circular nanopores on the Seebeck effect of graphene. a** Temperature-dependent Seebeck coefficient (*S*) and (**b**) *ZT* values of bilayer graphene with and without circular nanopores. Error bars represent standard deviations from three independent measurements. Density functional theory calculation of the band structure and projected density of state of bilayer graphene (**c**) without and (**d**) with ordered circular nanopores. Insets in (**c**) and (**d**) show the calculated Fermi energy surface. $\varepsilon_F$ represents the Fermi level. The bandgap ($E_g$) and zero density of state are indicated by the black and red double-headed arrows, respectively. The staggered, ordered nanoporous structure confines the electron transport by splitting the Dirac points and breaking up the Fermi energy surface, significantly enhancing the Seebeck effect of graphene. The diameter of the nanopores in the calculation was 1.2 nm.

Next, we sought to design an ordered porous graphene-based device to generate a temperature gradient upon EM radiation. We fabricated 25 mm long, 5 mm wide, and 10 µm thick graphene strips and coated one end of the strip with a 690 nm thick adiabatic parylene-c layer using CVD. Upon radiating the strips with EM waves with a frequency of 2.45 GHz using a homemade EM emitting cavity with a 100 W magnetron, the temperature profile and temperature gradient (defined as the temperature difference between the two ends of the graphene strip ($\Delta T$) divided by the strip length (25 mm)) as a function of time was determined using an infrared thermometer. As shown in Fig. 3h and Supplementary Figs. 21–22, the temperature gradient of the bilayer graphene with circular nanopores increases and reaches a maximum at 180 s with a value of 1.86 K/mm, which is approximately four times that of a nonporous bilayer graphene-based strip (0.52 K/mm). Upon removing the EM radiation, the temperature gradient returns to zero after 150 s. These results demonstrate that a large temperature gradient in ordered nanoporous graphene can be

generated upon EM radiation, further confirmed using finite-element analysis (Fig. 3i and Supplementary Fig. 23).

### Seebeck effect of nanoporous graphene

It is well known that pristine graphene lacks the strong Seebeck effect required for efficient DC generation from a temperature gradient[34,35]. We found that the presence of nanoporous structures greatly enhances the Seebeck effect of graphene, and the absolute value of the Seebeck coefficient ($|S|$) of ordered nanoporous graphene is comparable with conventional thermoelectric materials and larger than other modified graphene (Supplementary Table 5). As shown in Fig. 4a and Supplementary Fig. 24, the $|S|$ of bilayer graphene with ordered circular nanopores was between 69–83 µV/K over a temperature range of 300–500 K, at least six times larger than that of nonporous graphene (<10 µV/K), with the $|S|$ of monolayer and trilayer graphene being ~50% and ~15% smaller, respectively. The negative sign of *S* indicates an *n*-type semiconducting behavior for the ordered porous graphene, in which electrons are the predominant carriers[36]. We further evaluated the thermoelectric conversion ability of ordered porous graphene by calculating the largest dimensionless figure of merit, otherwise known as the *ZT* value ($ZT = \sigma S^2 T / \kappa_T$) and the thermoelectric coefficient[37]. As shown in Fig. 4b and Supplementary Fig. 24, we calculated the *ZT* value to be 0.33 over a temperature range of 300–500 K. We note that both the *ZT* value and the thermoelectric coefficient of the bilayer graphene with ordered circular nanopores are about three orders of magnitude higher than that of bilayer nonporous graphene, significantly higher than other graphene-based composites, and comparable to current state-of-the-art thermoelectric materials (Supplementary Tables 6 and 8).

To provide insight into the enhanced Seebeck effect of ordered porous graphene (high $|S|$ and *ZT* values), we performed temperature-dependent Hall carrier measurement analysis of ordered nanoporous graphene at different temperatures. As shown in Supplementary Fig. 13a-h and Supplementary Note 6, the presence of nanopores improves the effective mass and reduces the Hall carrier density. However, the $|S|$ value of a series of modified graphene with different effective masses and Hall carrier densities, including sulfur or nitrogen-doped graphene, graphene with polydisperse pore structures, and reduced graphene oxide, are less than 30 µV/K (Supplementary Fig. 25 and Supplementary Note 7). These results suggest that the synergistic effect of monodisperse nanometer-sized pores and the staggered porous structure alters the electronic structure and thus enhances the $|S|$ of graphene.

To elucidate the mechanisms that enhance $|S|$, we performed a first-principles density functional theory study of the band structure of the ordered nanoporous graphene (Supplementary Note 8). As evidenced in the band structure of bilayer nonporous graphene shown in Fig. 4c, the intrinsic zero bandgap electronic structure suggests a metallic characteristic of graphene that disfavors a high $|S|$ value. In contrast, the band structure of bilayer graphene with staggered circular pores (with an overlap ratio of 1/3) indicates that the Dirac points split and shift from their original positions (i.e., G, M, and K for nonporous graphene) to regions between G–M and K–G, resulting in an open bandgap structure with a bandgap of ~0.13 eV (shown in Fig. 4d). Furthermore, the splitting of the Dirac points leads to a breaking up of the original, continuous Fermi surfaces in the Brillouin zone into disconnected Fermi surfaces with different energies, which act as electron cages and confine carrier mobility in each Fermi surface (known as the Dirac trap effect)[38]. The calculated projected density of states in Fig. 4d reveals a high density of states (known as van Hoff singularity points) separated by zero density states. This electronic structure, combined with the Dirac trap effect, facilitates the interaction between electrons and phonons (known as the phonon drag effect) and tremendously enhances the $|S|$ of graphene[39]. Particularly, compared with bilayer graphene with completely overlapped nanopores (overlap

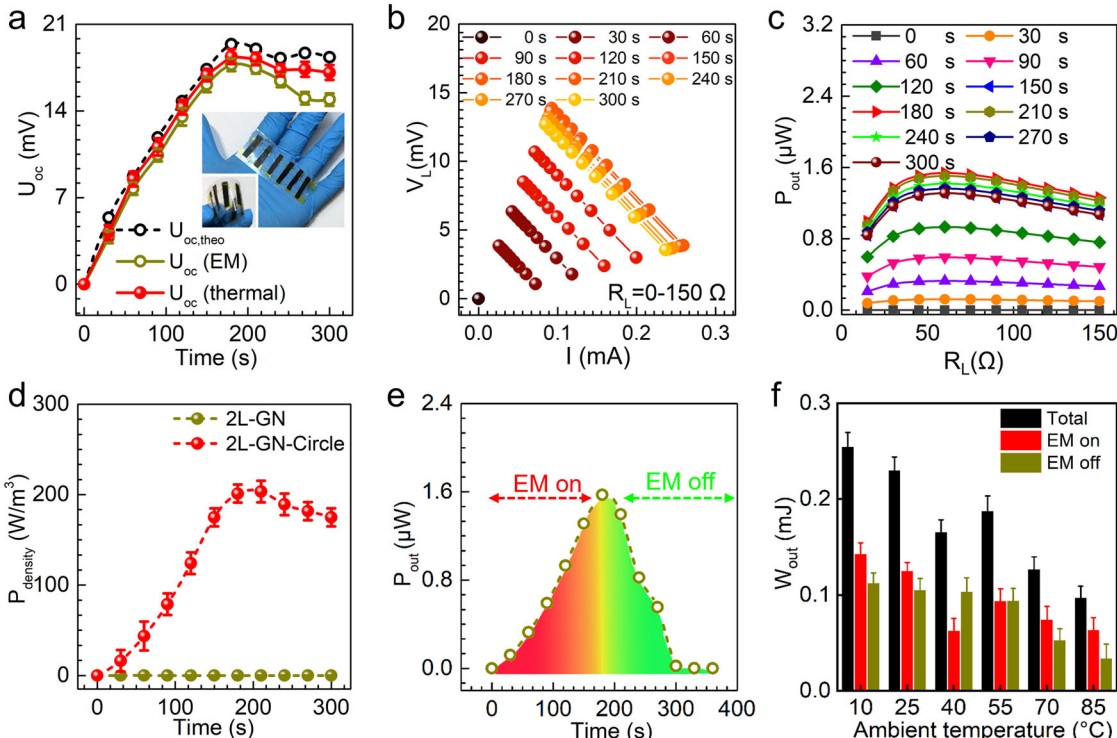

**Fig. 5 | Electric energy output performance of bilayer graphene with circular nanopores. a** Open-circuit voltage as a function of time. The EM radiation was applied from 0 to 180 s. The black and yellow curves represent experimental and theoretical open circuit voltage [$U_{oc\ (EM)}$ and $U_{oc,theo}$] under EM radiation, respectively. In the red curve displaying the maximum open circuit voltage from a temperature gradient [$U_{oc\ (thermal)}$], which is applied by using an external thermal source instead of EM radiation. Inset in (**a**) shows a photograph of the device assembled with six strips of porous bilayer graphene. **b** Loaded voltage ($V_L$) as a function of current ($I$) recorded via tuning the resistance from 0 to 150 Ω after different EM radiation times. **c** Output power ($P_{out}$) as a function of resistance load ($R_L$). **d** Power density ($P_{density}$) as a function of time. A typical 360 s-cycle includes a 180 s radiation period. **e** Output power ($P_{out}$) as a function of time at 25 °C. **f** Effect of temperature on the work output ($W_{out}$). The resistance load in (**e, f**) was 60 Ω. Error bars represent standard deviations from three independent measurements.

ratio of 1), the staggered pore structure with a partial overlap ratio facilitates the splitting of the Dirac points and thus generates higher intensity van Hoff singularity points, which are beneficial to the enhancement of the Seebeck effect of graphene (see Supplementary Figs. 26–28 for the effect of graphene layer number and overlap ratio)[40–42].

## Electricity output by nanoporous graphene

Based on this analysis, bilayer graphene with staggered circular nanopores possess the most desirable $\eta$, $\kappa_T$, and $|S|$ among the different nanoporous structures. In the final set of experiments, we fabricated a device consisting of six strips of ordered nanoporous graphene on an insulating polydimethylsiloxane substrate to explore its EM–heat–DC conversion. As shown in Fig. 5a, the open-circuit voltage ($U_{oc}$) of the graphene-based device was measured to increase to ~17.7 mV over about 180 s before decreasing with continued EM radiation. In contrast, the $U_{oc}$ of a nonporous graphene-based device was only 0.8% of that (Supplementary Fig. 29). We calculated the theoretical $U_{oc,theo}$ as[43]:

$$U_{oc,theo} = \triangle T \cdot N \cdot |S| \qquad (3)$$

where $N$ is the number of strips of ordered porous graphene and the average $|S|$ of the bilayer porous graphene between 300–450 K was 72.3 μV/K. The experimental $U_{oc}$ of the ordered porous graphene is 71.0–92.8% of the $U_{oc,theo}$. We hypothesize that this difference between $U_{oc}$ and $U_{oc,theo}$ is caused by the interference between the EM and thermal fields – the mismatch between the direction of the alternating EM field and the temperature gradient causes a deviation of the flux of electrons and thus lowers $U_{oc}$. This hypothesis is validated by directly

applying the same $\Delta T$ to the porous graphene-based device without EM radiation. The measured $U_{oc}$, in this case, is in better quantitative agreement with $U_{oc,theo}$. The effect of this interference on graphene with different pore structures is discussed in Supplementary Note 9 and Supplementary Fig. 30.

By characterizing the output voltage–current curve of the ordered nanoporous graphene-based devices as a function of load resistance (0–150 Ω) and EM radiation time (Fig. 5b), we measured the maximum output power and power density to be ~1.5 μW and ~204 W/m³ with a load resistance of 60 Ω upon EM radiation for 180 s, as shown in Fig. 5c, d. We note here that the maximum power density is several times that of conventional thermoelectric materials (Supplementary Table 7). We also note that the device's output power can be enhanced by increasing the thickness and number of the porous graphene strips (Supplementary Note 10 and Supplementary Fig. 31). The device can output electric energy of ~0.23 mJ per EM radiation cycle (180 s on and 180 s off; Fig. 5d) for at least 500 cycles. The overall EM–electricity conversion of the ordered nanoporous graphene-based device is calculated to be ~5.6% (Supplementary Note 11), which is attributed to the high EM dissipation factor and high Seebeck coefficient. In addition, the thermoelectric coefficient of the device is ~14.4%, which is at least three orders of magnitude larger than that of pristine graphene and comparable to that of state-of-the-art thermoelectric material-based devices (Supplementary Table 8). Finally, the ordered nanoporous graphene-based devices exhibited an excellent electricity generation performance over a wide temperature range (10–85 °C; Fig. 5f and Supplementary Fig. 32), even after being bent 500 times (Supplementary Fig. 33).

In this work, we report a facile method to create staggered, monodisperse nanometer-sized circular pores on the surface of

graphene. It was observed that the presence of these nanopores and the staggered porous structure alter the electronic and phononic structure of the graphene (e.g., dipole polarization relaxation, open bandgap structure, splitting of Dirac points, and phonon scattering), resulting in desirable functionalities, such as a high permittivity, low thermal conductivity, and high Seebeck coefficient and *ZT* values. As a result, the ordered nanoporous graphene enables the efficient generation of DC electricity through the absorption and dissipation of EM waves into heat, followed by a simultaneous thermoelectric process. Our results advance the fundamental understanding of the structure–property relationship of ordered nanoporous graphene and provide a design strategy to efficiently absorb and convert low-frequency EM waves and waste heat from the environment into usable electricity. This class of multifunctional ordered nanoporous graphene may expand the potential utility of graphene in energy harvesting, thermal management, EM shielding and absorption, thermoelectrics, and photocatalytic water splitting. Future efforts will seek to experimentally control the overlap ratio of nanopores on different graphene layers to study its influence on the graphene properties. In addition, the self-powering and self-charging of wearable electronics, building on the success of emerging 5G communications, using ordered porous graphene-based devices are being investigated.

## Methods

### Materials

Iron acetylacetonate, oleic acid, oleyl amine, ammonia (28 wt %), hydrogen peroxide (30 wt %), sodium molybdate dehydrate, thiourea, ferrous chloride, cobalt acetate, ethylene glycol, zinc acetate, lanthanum oxide, cobalt nitrate, tetraethyl orthosilicate, sodium hydroxide, oxalic acid, strontium carbonate, cobalt oxalates, bismuth nitrate pentahydrate, tellurium powder, 1-butyl-3-methylimidazolium bromide, silver nitrate, stannous chloride, sulfur powder, ethylenediaminetetraacetic acid disodium, copper nitrate, hexadecyltrimethylammonium bromide, formaldehyde, cyclohexane, isopropanol, iron pentacarbonyl, 1,3-dimethylimidazoline-2-selenone, hydrochloric acid (38 wt %), and parylene-c monomer were bought from Sigma-Aldrich. Raw graphene and multiwall carbon nanotubes were obtained from XF-NANO tech. Co. All chemical reagents are analytical pure reagents and were used without further purification. Polydimethylsiloxane film was bought from Shanghai Muke Technol. Co. Water used in all the experiments was purified using a Milli-Q water purification system (Simplicity C9210).

### Synthesis of graphene with ordered nanometer-sized pores

First, iron acetylacetonate, raw graphene, prepared by chemical vapor deposition (CVD), oleic acid, and oleyl amine were mixed and ultrasonicated for 30 min. Afterward, the mixture was heated between 110 and 120 °C with vigorous stirring for 2 h under a nitrogen flow. Next, the mixture was heated to 220 °C and kept at this temperature for 30 min, followed by heating up to 300 °C at a heating rate of 2 °C/min where it stayed for another 30 min. After cooling to room temperature, the graphene decorated with iron oxide nanoparticles was collected via centrifugation after dispersing it in cyclohexane and isopropanol.

The iron oxide nanoparticle-coated graphene was kept at 300 °C for 60 min under a constant nitrogen flow to remove residual organics. Next, the temperature was increased to 900 °C at a rate of 1 °C/min and maintained at this temperature for 2 h to create nanopores on the graphene. After cooling to room temperature, the graphene was dispersed into a hydrochloric acid solution with a pH between 1.5 and 2.0 to remove the residual nanoparticles, resulting in the formation of nanometer-sized pores on the graphene surface. The nanopore shapes can be tuned by forming nanoparticles with different geometries (see Supplementary Table 9). By utilizing raw graphene with specific numbers of layers, we can fabricate double layer, trilayer, or multilayer graphene with nanopores.

### Synthesis of graphene with nanopore sizes less than 3 nm

In total, 0.1 g of the bilayer graphene prepared by CVD was added to 100 mL of hydrogen peroxide ($H_2O_2$; 30 wt %) and was ultrasonicated for 30 min. Afterward, the solution was heated to 80 °C and kept at this temperature for 1.2 h under magnetic stirring. Finally, the nanoporous graphene with pore diameters <3 nm was collected using centrifugation and dried at 80 °C for 24 h.

### Synthesis of nitrogen (N)-doped graphene

10 mg of bilayer graphene prepared by CVD was dispersed in 50 mL of distilled water and ultrasonicated for 30 min. Next, 0.8 mL of pyrrole monomer was dissolved in the graphene dispersion and ultrasonicated for another 30 min. The mixture was transferred to an autoclave and heated at 180 °C for 12 h. The N-doped graphene was collected and washed using distilled water, followed by a vacuum drying process at 60 °C for 24 h.

### Synthesis of sulfur (S)-doped graphene

100 mg of thioacetamide and 10 mg of bilayer graphene prepared by CVD were dispersed into a solvent mixture consisting of 30 mL of distilled water, 0.1 mL of $H_2O_2$ (30 wt %), and 0.1 mL of hydrochloric acid (HCl; 38 wt %). Next, the solution was transferred to an autoclave and heated at 150 °C for 10 h. The S-doped graphene was washed using distilled water and dried at 60 °C for 24 h under a vacuum.

### Fabrication of graphene-based devices for EM–heat–DC conversion

A thin film of ordered nanoporous graphene with a thickness of ~10.0 ± 0.8 μm was spin-coated on a silicon substrate and later cut into several 25 mm long and 5 mm wide strips. Next, 20 mm of each strip was covered, and the exposed region (5 mm) of the graphene was coated with parylene-c using a CVD process involving four steps at 120, 130, 140, and 150 °C with corresponding times of 20, 20, 40, and 0–120 min, respectively. The temperature of the CVD deposition chamber was set to 650 °C, and the parylene-c monomer was placed in the CVD evaporation chamber set at 120 °C. The thickness of the parylene-c coating was tuned by the chemical deposition time. After deposition, six parylene-c-coated graphene strips were assembled onto a 20 μm-thick polydimethylsiloxane film, with ~5 mm spacing between each strip. Finally, all the strips were connected in series using conductive copper paste.

### Structural characterization

The phase identification of graphene materials was performed using a Bruker D8 ADVANCE X-ray diffractometer (XRD) with Cu Kα radiation (λ = 0.15406 nm). A Joel JEM 2100 F transmission electron microscope (TEM) with a 200 kV field emission was employed to investigate the morphology of the nanoparticles and graphene. X-ray photoelectron spectroscopy analyses were performed using an Escalab 250Xi spectrometer. The graphitization level of the ordered nanoporous graphene was measured using a Jobin Yvon HR 800 confocal Raman spectroscope.

### Evaluation of the EM dissipation ability of graphene

The temperature-dependent permittivity of graphene was measured using an Agilent N5232 vector network analyzer based on the single-port transmission-line theory. Specifically, a 30 vol % mixture of graphene in silicone resin was pressed into a toroidal ring with an outer diameter of 7.00 mm and an inner diameter of 3.04 mm. The permittivity of the graphene and silicone resin mixture can be calculated as[44]

$$\log_{10}\varepsilon_{mixture} = \lambda_{graphene}\log_{10}\varepsilon_{graphene} + \lambda_{silicon}\log_{10}\varepsilon_{silicon} \qquad (4)$$

where $y_{graphene}$ and $y_{silicon}$ represent the volume fraction of the graphene and silicone resin, respectively. $\varepsilon_{mixture}$, $\varepsilon_{graphene}$, and $\varepsilon_{silicon}$

represent the permittivity of the silicone resin/graphene mixture, pure graphene, and pure silicone resin, respectively. After $\varepsilon_{silicon}$ was measured, $\varepsilon_{graphene}$ was calculated based on Eq. (4). We note here that the uncertainty in the permittivity measurements is about 10–12% due to variations originating from sample preparation.

**Characterization of the thermal conductivity, electric conductivity, and Seebeck coefficient of graphene**

The electric conductivity ($\sigma$) of graphene materials was measured according to a four-probe method using a Keithley 4200-SCS electrometer. The thickness of the graphene film was measured using a Tencor profilometer. The surface morphology of the graphene sample was characterized using a Park NX10 atomic force microscope. The graphene's temperature-dependent $\sigma$ and Seebeck coefficient ($S$) were measured using an Ulvac-Riko ZEM-3 system. The graphene was hot-pressed into a pellet with a length of ~12 mm and a width of ~3 mm. The thermal conductivity ($\kappa_T$) of graphene can be calculated as[45]:

$$\kappa_T = D C_p \lambda \quad (5)$$

where $D$ is the density, $C_p$ is the specific heat value, and $\lambda$ is the thermal diffusivity. $\kappa_T$ was measured using a Netzsch LFA-467 laser flash system according to the laser flash diffusivity method with a pyroceram disk as reference. In addition, the specific heat values were collected using a Pyroceram 9606 system. Hall carrier density was measured using a physical property measurement system. We note here that the uncertainty of the electrical output value is about 15–25% due to the error in the measurement of $\sigma$, $\kappa_T$, and $S$. The error of the temperature is around 1 K.

## Data availability

Relevant data supporting the key findings of this study are available within the article and the Supplementary Information file. All raw data generated during the current study are available from the corresponding authors upon request.

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

## Acknowledgements

X. Wang would like to acknowledge the financial support from the startup fund of the Ohio State University (OSU), OSU Sustainability Institute Seed Grant, and OSU Institute for Materials Research Kickstart Facility Grant.

## Author contributions

H.L., R.C., R.W., S.A. and X.W. designed the experiments, supervised the experiments, and wrote the manuscript. H.L., Y.Y., S.L., G.W., R.L.D., U.I.K., Y.Z., S.X., X.Z. and H.X. conducted material synthesis and performed electron microscopy, XRD, XAFS, Raman characterization, and electromagnetic analysis. G.W., B.Z., H.X. and J.Z. carried out the thermal conductivity, Hall curves, and Seebeck characterization. H.L. and B.L. performed molecular dynamics simulation. H.L. and J.Z. performed the first-principles density functional theory simulation and finite-element analysis. All the authors discussed the results and contributed to the manuscript preparation. H.L., Y.Y., S.L., G.W. and B.Z. contributed equally to this work.

## Competing interests

The authors declare no competing interests.

## Additional information

**Peer review information** : *Nature Communications* thanks Sebastian Volz and the other, anonymous, reviewer(s) for their contribution to the peer review of this work. Peer reviewer reports are available.

