## [Peer Review File · Nature Communications]

Staggered circular nanoporous graphene converts electromagnetic waves into electricityREVIEWER COMMENTS

Reviewer #1 (Remarks to the Author):

The authors proposed the new energy harvester using an electromagnetic wave-heat-electricity energy conversion material. The data is very solid and paper has been well rewritten. The motivation of this paper is of significant and valuable that focusing on the 5G low-frequency band instead of other high frequency region. Furthermore, the most impressive me that the succussive fabrication of the ordered nanometer-sized porous graphene with fine-controlled the pore shapes and pore-staggering. Meanwhile, interesting physical findings such as low-frequency relaxation polarization, ultralow thermal conductivity and Seebeck, new insights have been well analyzed via systematical evidence.

Overall, this paper advances the fundamental understanding of the structure–property relationships of ordered nanoporous graphene, and attractive to broad readers, not limiting to EM field.

I recommend this paper to Nat. Commun.

Of course, some suggestions should be concerned also:

1: In the SI paper, Fig S11-17 can be deleted because the author has listed a table to compare the performance with current reports.

2: As we know, to obtain a high EM dissipation performance, the low-frequency relaxation behavior is very important. Why graphene could not induce lower-frequency relaxation behavior at low-frequency region?

3: How about the thermoelectric performance of graphene? Such as the ZT value?

4: English should be polished further to avoid mistakes.

Reviewer #2 (Remarks to the Author):

An original process allows for introducing nanopores with oxydized boundaries in few layer graphene. The resulting film displays enhanced GHz EM energy absorption and thermoelectric (TE) properties.

The combination of both is proposed to harvest EM noise and transduce it into DC electrical current.

The obtained high dielectric loss tangent, low thermal conductivity and high Seebeck finally yield an output power of 1.6 microW for a several cms long ribbon.

The technological strategy (EMI harvesting) is smart and timely, however, the generation of heat in a device is detrimental and the gain is far below the efficiencies of today's equivalent TE harvesters.

On the scientific point of view, it is well-known in theory that TE properties should be improved in nanostructured graphene. To their benefit, authors provide an experimental demonstration. Comparisons with other state-of-the-art graphene based converters remain to be shown. The TE coefficient of performance and ZT should have also been provided.

The improvement of the dielectric loss tangent in the 5G frequency range seems to be the most impacting outcome. An efficient ultra-thin EM shield is certainly of technological value. It seems reasonable to claim that dipole relaxation on hole boundaries is the relevant physical mechanism.

It is to note that several key effects are not properly cited ('dipole relaxation polarization', 'dielectric tangent loss'), which yields a strange feeling of inaccuracy.

To conclude, this work appears more as an engineering project consisting of an

interesting optimization of a few-layer graphene film for EM absorption and a more conventional, rather poorly efficient, TE conversion.

The aggregation of numerous studies provided by a large consortium and leading to a plethora of results (more than 100 figures) does not here produce a larger scientific impact.

It finally seems -to fit the format of a paper- that this article should either be reduced to its most exciting part or submitted to a nano / engineering journal.

Reviewer #3 (Remarks to the Author):

The authors proposed an energy harvesting material from electromagnetic waves to electricity by using Graphene. The graphene is heated by induction, then electricity is generated by thermoelectric effects. The porous graphene shows a higher performance than the normal graphene due to enhanced thermoelectric properties. It is interesting. However, the generated power is 1.5 microW under the 1000W electromagnetic wave irradiation. If the proposed graphene is used for energy harvesting, the irradiated power is in the order of 10mW at most. Only 1nW generated power might be too low for practical use even if it is for energy harvesting.

Reviewer 1:

The authors proposed the new energy harvester using an electromagnetic wave-heat-electricity energy conversion material. The data is very solid and paper has been well rewritten. The motivation of this paper is of significant and valuable that focusing on the 5G low-frequency band instead of other high frequency region. Furthermore, the most impressive me that the succussive fabrication of the ordered nanometer-sized porous graphene with fine-controlled the pore shapes and pore-staggering. Meanwhile, interesting physical findings such as low-frequency relaxation polarization, ultralow thermal conductivity and Seebeck, new insights have been well analyzed via systematical evidence.

Overall, this paper advances the fundamental understanding of the structure–property relationships of ordered nanoporous graphene, and attractive to broad readers, not limiting to EM field.

I recommend this paper to Nat. Commun.

Of course, some suggestions should be concerned also:

Response: We thank the reviewer for their careful reading of our manuscript and their positive comments regarding the significance and novelty of our work.

1: In the SI paper, Fig S11-17 can be deleted because the author has listed a table to compare the performance with current reports.

Response: We thank the reviewer for their detailed comment and careful reading. To highlight the most important findings of this work, we have significantly shortened the Supplementary Information by reducing its length from 113 pages to 60 pages and the number of Supplementary Figures from 67 to 34 (including removing the original Supplementary Figures 11-17). Accordingly, we refer the reviewer to our revised Supplementary Information.

2: As we know, to obtain a high EM dissipation performance, the low-frequency relaxation behavior is very important. Why graphene could not induce lower-frequency relaxation behavior at low-frequency region?

Response: Pristine graphene, especially that fabricated by chemical vapor deposition (CVD), is typically of a high quality with a limited number of defects, which gives rise to insufficient dipoles and no significant relaxation behavior in response to GHz-frequency EM fields. To clarify this point, we have added the following text to the revised manuscript:

“Pristine graphene fabricated by chemical vapor deposition (CVD) possesses a limited number of defects, and thus lacks sufficient dipoles, which largely restricts the relaxation behavior in response to GHz-frequency EM fields.”

3: *How about the thermoelectric performance of graphene? Such as the ZT value?*

Response: To address the reviewer's comment, we have calculated the ZT value for ordered nanoporous graphene using $ZT = \sigma S^2 T / \kappa_T$, where S , σ , T , and κ_T are the Seebeck coefficient, electric conductivity, absolute temperature, and thermal conductivity, respectively. In comparison to other materials, the ZT of bilayer graphene with circular nanopores is at least two orders of magnitude larger than that of pristine and defective graphene and even comparable with conventional thermoelectric materials. We have added the following text to the revised manuscript:

“We further evaluated the thermoelectric conversion ability of ordered porous graphene by calculating the ZT value ($ZT = \sigma S^2 T / \kappa_T$)³⁷ and the thermoelectric coefficient³⁷. As shown in **Fig. 4b** and **Supplementary Fig. 26**, we calculated the largest dimensionless figure of merit ZT ($ZT = S^2 \sigma T / \kappa_T$) to be 0.33 over a temperature range of 300–500 K. We note that both the ZT value and the thermoelectric coefficient of the bilayer graphene with ordered circular nanopores are about three orders of magnitude higher than that of bilayer nonporous graphene, significantly higher than other graphene-based composites, and comparable to current state-of-the-art thermoelectric materials (**Supplementary Tables 6 and 8**).”

We have added Figure 4b to the revised manuscript:

Fig. 4. Effect of staggered, ordered nanopores on the Seebeck effect of graphene. (a) Temperature-dependent Seebeck coefficient (S) and **(b) ZT value** of bilayer graphene with and without circular nanopores. Error bars represent standard deviations from three independent measurements.”

We have added Supplementary Figure 26, Supplementary Table 6 and the following references to the revised Supplementary Information:

Supplementary Fig. 26 - Temperature-dependent Seebeck coefficient (S) and ZT value of graphene with different numbers of layers and pore shapes. (a-d) S and (e-h) ZT value of porous graphene with (a, e) 1, (b, f) 2, (c, g) 3, and (d, h) 6 layers as a function of temperature.

Supplementary Table 6. ZT values of conventional thermoelectric materials and graphene.

Sample	Maximum ZT (Temperature range)	ZT at 500K	Ref.
Single layer graphene	<0.01 at 600K (300K-600K)	<0.01	108
Defective graphene	<0.01 at 300K (300K)	N.A.	109
Graphene/ Bi_2Te_3	0.55 at 500K (300K-500K)	0.55	110
Graphene/ CoSb_3	0.61 at 800K (300K-800K)	~0.15	111
Graphene/ $\text{Bi}_{1-x}\text{Te}_3$	0.2 at 480K (300K-480K)	N.A.	112
Graphene/ CuInTe_2	0.4 at 700K (300K-700K)	0.1	113
Cu_2Se	1.2 at 1000K (300K-1000K)	~0.1	114
AgMnSbTe_3	1.4 at 800K (300K-800K)	~0.6	115
Ga-doped PbTe	1.55 at 723K (300K-723K)	~0.5	116
AgMnGeSbTe_4	1.05 at 773K (300K-773K)	~0.6	117
$\text{Pb}_7\text{Bi}_4\text{Se}_{13}$	1.35 at 800K (300K-800K)	~0.5	118
(Pb, Ge, Sb, Cd)-doped SnTe	1.5 at 800K (300K-800K)	~0.6	119

Cu ₁₂ Sb ₄ S ₁₃ -based alloy	1.15 at 723K (300K-723K)	0.4	120
Cu ₂ SnSe ₃	1.6 at 823K (300K-823K)	<0.2	121
Bi ₂ Si ₂ Te ₆	0.51 at 623K (300K-623K)	<0.3	122
ZrRu _{1+x} Sb	0.2 at 973K (300K-973K)	<0.1	123
2L-GN-Circle	0.33 at 500K (300K-500K)	0.33	This work

108. Reshak, A. *et al.* Thermoelectric properties of a single graphene sheet and its derivatives. *J. Mater. Chem. C* **2**, 2356 (2014).
109. Anno, Y. *et al.* Enhancement of graphene thermoelectric performance through defect engineering. *2D Mater.* **4**, 025019 (2017).
110. Ahmad, K. *et al.* Enhanced thermoelectric performance of Bi₂Te₃ based graphene nanocomposites. *Appl. Surf. Sci.* **2**, 474 (2019).
111. Feng, B. *et al.* Enhanced thermoelectric properties of *p*-type CoSb₃/graphene nanocomposite. *J. Mater. Chem. A* **1**, 13111 (2013).
112. Liang, B. B. *et al.* Fabrication and thermoelectric properties of graphene/Bi₂Te₃ composite materials. *J. Nanomater.* **6**, 6 (2013).
113. Chen, H.J. *et al.* Thermoelectric properties of CuInTe₂/graphene composites. *CrystEngComm* **15**, 6648 (2013).
114. Choo, S. *et al.* Cu₂Se-based thermoelectric cellular architectures for efficient and durable power generation. *Nat. Commun.* **12**, 3550 (2021).
115. Luo, Y.B. *et al.* Cubic AgMnSbTe₃ semiconductor with a high thermoelectric performance. *J. Am. Chem. Soc.* **143**, 13990 (2021).
116. Luo, Z.Z. *et al.* Extraordinary role of Zn in enhancing thermoelectric performance of G-doped *n*-type PbTe. *Energy Environ. Sci.* **15**, 368 (2022).
117. Ma, Z. *et al.* High entropy semiconductor AgMnGeSbTe₄ with desirable thermoelectric performance. *Adv. Funct. Mater.* **21**, 2103197 (2013).
118. Hu, L. *et al.* High thermoelectric performance enabled by convergence of nested conduction bands in Pb₇Bi₄Se₁₃ with low thermal conductivity. *Nat. Commun.* **12**, 4793 (2021).
119. Zhang, Q. *et al.* High-performance thermoelectric material and module driven by medium-entropy engineering in SnTe. *Adv. Funct. Mater.* **32**, 220548 (2022).
120. Hu, H.H. *et al.* Thermoelectric Cu₁₂Sb₄S₁₃-based synthetic minerals with a sublimation-derived porous network. *Adv. Mater.* **33**, 2103633 (2021).
121. Hu, L. *et al.* High thermoelectric performance through crystal symmetry enhancement in triply doped diamondoid compound Cu₂SnSe₃. *Adv. Energy Mater.* **11**, 2100661 (2021).
122. Luo, Y. B. *et al.* Thermoelectric performance of 2D Bi₂Si₂Te₆ Semiconductor. *J. Am. Chem. Soc.* **144**, 1445 (2022).
123. Wang, L.Y. *et al.* Discovery of a slater-pauling semiconductor ZrRu_{1.5}Sb with promising thermoelectric properties. *Adv. Funct. Mater.* **32**, 2200438 (2022).”

4: English should be polished further to avoid mistakes.

Response: We thank the reviewer for carefully reading our manuscript. The manuscript has been proofread throughout to ensure the clarity and accuracy of the writing.

Reviewer 2:

An original process allows for introducing nanopores with oxidized boundaries in few layer graphene. The resulting film displays enhanced GHz EM energy absorption and thermoelectric (TE) properties. The combination of both is proposed to harvest EM noise and transduce it into DC electrical current.

The obtained high dielectric loss tangent, low thermal conductivity and high Seebeck finally yield an output power of 1.6 microW for a several cms long ribbon.

The technological strategy (EMI harvesting) is smart and timely, however, the generation of heat in a device is detrimental and the gain is far below the efficiencies of today's equivalent TE harvesters.

Response: We thank the reviewer for carefully reading our manuscript and the positive comments regarding our “*smart and timely*” technological strategy. To address the reviewer’s concern about the EM–electricity conversion from our ordered porous graphene-based device, we estimated the overall EM–electricity conversion of our device using the following equation:

$$x = \frac{W_{\text{out}}}{\frac{V_{\text{device}}}{V_{\text{chamber}}} W_{\text{source}} \alpha}$$

where W_{out} is the output work of the graphene device (0.23 mJ during 180 s of EM radiation), W_{source} is the output work of the EM generator, α is the attenuation degree, and V_{device} and V_{chamber} represent the volume of the graphene device and EM radiation chamber, respectively. Note that we should not use the total of the output electric power to calculate the overall EM–electricity conversion, as only a small fraction of the EM waves is absorbed by the device. Due to the large wavelength of EM, its wave attenuation is negligible within the chamber ($\alpha \approx 1$), leading to an almost constant EM intensity in the chamber (see *IEEE Trans. Mol. Biol. Multi-Scale Commun.* 2015, 1, 18 and *Appl. Phys. Lett.* 2001, 78, 16). Therefore, the fraction of the EM waves exposed to the graphene device can be calculated by the volume fraction of the graphene device to the volume of the chamber. The overall EM–electricity conversion of our graphene device is calculated to be $\sim 5.6\%$. To clarify this point, we have added the following text to the revised manuscript:

“The overall EM–electricity conversion of porous bilayer graphene is estimated to be $\sim 5.6\%$ (Supplementary Note 10).”

We have also added Supplementary Note 10 to the revised Supplementary Information:

“Supplementary Note 10. Calculation of the overall EM–electricity conversion of ordered nanoporous graphene-based devices

The overall EM–electricity conversion (x) of our ordered porous graphene-based device can be calculated as:

$$x = \frac{W_{\text{out}}}{\frac{V_{\text{device}}}{V_{\text{chamber}}} W_{\text{source}} \alpha} \quad (13)$$

where W_{out} is the output work of the graphene device (0.23 mJ during 180 s of EM radiation), W_{source} is the output work of the EM generator (100 W), α is the attenuation degree, and V_{device} and V_{chamber} represent the volume of the graphene device (six graphene strips; each strip is 25 mm \times 5 mm \times 0.01 mm) and EM radiation chamber (447 mm \times 247 mm \times 281 mm), respectively. It is well established that the GHz-frequency EM wave attenuation is negligible in a chamber with these dimensions ($\alpha \approx 1$), which gives rise to an almost constant EM intensity in the chamber^{31,32}. Therefore, the fraction of the EM waves exposed to the graphene device can be calculated by the volume fraction of the graphene device relative to the chamber volume. As a result, the overall EM–electricity conversion of the bilayer porous graphene-based device was calculated to be 5.6 %.”

In addition, we have added the following references to the revised Supplementary Information:

“31. Toyoshima, M. *et al.* Comparison of microwave and light wave communication systems in space applications. *Opt. Eng.* **47**, 015003 (2007).

32. Heremans J. P. *et al.* Enhancement of thermoelectric efficiency in PbTe by distortion of the electronic density of states. *Science* **321**, 554 (2008).”

On the scientific point of view, it is well-known in theory that TE properties should be improved in nanostructured graphene. To their benefit, authors provide an experimental demonstration. Comparisons with other state-of-the-art graphene based converters remain to be shown. The TE coefficient of performance and ZT should have also been provided.

Response: To address the reviewer’s comments regarding the ZT value and thermoelectric coefficient, we have calculated the ZT value of the ordered nanoporous graphene using $ZT = \sigma S^2 T / \kappa_T$, where S , σ , T , and κ_T are the Seebeck coefficient, electric conductivity, absolute temperature, and thermal conductivity, respectively. In comparison to other materials, the ZT of bilayer graphene with circular nanopores is 0.33, at least two orders of magnitude larger than that of pristine and defective graphene and even comparable with conventional thermoelectric materials.

Furthermore, we have calculated the thermoelectric coefficient (C_{TE}) of ordered porous graphene as:

$$C_{\text{TE}} = \frac{T_h - T_c}{T_h} \frac{\sqrt{1 + \overline{ZT}} - 1}{\sqrt{1 + \overline{ZT} + T_c/T_h}}$$

where T_h and T_c are the temperature of the hot and cold ends ($\Delta T = T_h - T_c$), respectively. \overline{ZT} is the average ZT over the temperature range, which is calculated by the integration method (i.e., a fourth-order polynomial fitting of the ZT curve). The thermoelectric coefficient of the bilayer porous graphene-based device (six strips) is $\sim 14.4\%$, with each strip at $\sim 2.4\%$. We note that the thermoelectric coefficient of bilayer graphene with ordered circular nanopores is at least three orders of magnitude larger than that of pristine graphene prepared by CVD and reduced graphene oxide and even comparable with conventional thermoelectric materials. We have added the following text to the revised manuscript:

“We further evaluated the thermoelectric conversion ability of ordered porous graphene by calculating the ZT value ($ZT = \sigma S^2 T / \kappa_T$)³⁷ and the thermoelectric coefficient³⁷. As shown in **Fig. 4b** and **Supplementary Fig. 26**, we calculated the largest dimensionless figure of merit ZT ($ZT = S^2 \sigma T / \kappa_T$) to be 0.33 over a temperature range of 300–500 K. We note that both the ZT value and the thermoelectric coefficient of the bilayer graphene with ordered circular nanopores are about three orders of magnitude higher than that of bilayer nonporous graphene, significantly higher than other graphene-based composite, and comparable to current state-of-the-art thermoelectric materials (**Supplementary Tables 6 and 8**).”

We have added Figure 4b to the revised manuscript:

Fig. 4. Effect of staggered, ordered nanopores on the Seebeck effect of graphene. (a) Temperature-dependent Seebeck coefficient (S) and **(b) ZT value** of bilayer graphene with and without circular nanopores. Error bars represent standard deviations from three independent measurements.”

We have added Supplementary Figure 26, Supplementary Tables 6 and 8, and the following references to the revised Supplementary Information:

Supplementary Fig. 26 - Temperature-dependent Seebeck coefficient (S) and ZT value of graphene with different numbers of layers and pore shapes. (a-d) S and (e-h) ZT value of porous graphene with (a, e) 1, (b, f) 2, (c, g) 3, and (d, h) 6 layers as a function of temperature.

Supplementary Table 6. ZT values of conventional thermoelectric materials and graphene.

Sample	Maximum ZT (Temperature range)	ZT at 500K	Ref.
Single layer graphene	<0.01 at 600K (300K-600K)	<0.01	108
Defective graphene	<0.01 at 300K (300K)	N.A.	109
Graphene/ Bi_2Te_3	0.55 at 500K (300K-500K)	0.55	110
Graphene/ CoSb_3	0.61 at 800K (300K-800K)	~0.15	111
Graphene/ $\text{Bi}_{1-x}\text{Te}_3$	0.2 at 480K (300K-480K)	N.A.	112
Graphene/ CuInTe_2	0.4 at 700K (300K-700K)	0.1	113
Cu_2Se	1.2 at 1000K (300K-1000K)	~0.1	114
AgMnSbTe_3	1.4 at 800K (300K-800K)	~0.6	115
Ga-doped PbTe	1.55 at 723K (300K-723K)	~0.5	116
AgMnGeSbTe_4	1.05 at 773K (300K-773K)	~0.6	117
$\text{Pb}_7\text{Bi}_4\text{Se}_{13}$	1.35 at 800K (300K-800K)	~0.5	118
(Pb, Ge, Sb, Cd)-doped SnTe	1.5 at 800K (300K-800K)	~0.6	119

Cu ₁₂ Sb ₄ S ₁₃ -based alloy	1.15 at 723K (300K-723K)	0.4	120
Cu ₂ SnSe ₃	1.6 at 823K (300K-823K)	<0.2	121
Bi ₂ Si ₂ Te ₆	0.51 at 623K (300K-623K)	<0.3	122
ZrRu _{1+x} Sb	0.2 at 973K (300K-973K)	<0.1	123
2L-GN-Circle	0.33 at 500K (300K-500K)	0.33	This work

108. Reshak, A. *et al.* Thermoelectric properties of a single graphene sheet and its derivatives. *J. Mater. Chem. C* **2**, 2356 (2014).
109. Anno, Y. *et al.* Enhancement of graphene thermoelectric performance through defect engineering. *2D Mater.* **4**, 025019 (2017).
110. Ahmad, K. *et al.* Enhanced thermoelectric performance of Bi₂Te₃ based graphene nanocomposites. *Appl. Surf. Sci.* **2**, 474 (2019).
111. Feng, B. *et al.* Enhanced thermoelectric properties of *p*-type CoSb₃/graphene nanocomposite. *J. Mater. Chem. A* **1**, 13111 (2013).
112. Liang, B. B. *et al.* Fabrication and thermoelectric properties of graphene/Bi₂Te₃ composite materials. *J. Nanomater.* **6**, 6 (2013).
113. Chen, H.J. *et al.* Thermoelectric properties of CuInTe₂/graphene composites. *CrystEngComm* **15**, 6648 (2013).
114. Choo, S. *et al.* Cu₂Se-based thermoelectric cellular architectures for efficient and durable power generation. *Nat. Commun.* **12**, 3550 (2021).
115. Luo, Y.B. *et al.* Cubic AgMnSbTe₃ semiconductor with a high thermoelectric performance. *J. Am. Chem. Soc.* **143**, 13990 (2021).
116. Luo, Z.Z. *et al.* Extraordinary role of Zn in enhancing thermoelectric performance of Ga-doped n-type PbTe. *Energy Environ. Sci.* **15**, 368 (2022).
117. Ma, Z. *et al.* High entropy semiconductor AgMnGeSbTe₄ with desirable thermoelectric performance. *Adv. Funct. Mater.* **21**, 2103197 (2013).
118. Hu, L. *et al.* High thermoelectric performance enabled by convergence of nested conduction bands in Pb₇Bi₄Se₁₃ with low thermal conductivity. *Nat. Commun.* **12**, 4793 (2021).
119. Zhang, Q. *et al.* High-performance thermoelectric material and module driven by medium-entropy engineering in SnTe. *Adv. Funct. Mater.* **32**, 220548 (2022).
120. Hu, H.H. *et al.* Thermoelectric Cu₁₂Sb₄S₁₃-based synthetic minerals with a sublimation-derived porous network. *Adv. Mater.* **33**, 2103633 (2021).
121. Hu, L. *et al.* High thermoelectric performance through crystal symmetry enhancement in triply doped diamondoid compound Cu₂SnSe₃. *Adv. Energy Mater.* **11**, 2100661 (2021).
122. Luo, Y. B. *et al.* Thermoelectric performance of 2D Bi₂Si₂Te₆ Semiconductor. *J. Am. Chem. Soc.* **144**, 1445 (2022).
123. Wang, L.Y. *et al.* Discovery of a slater-pauling semiconductor ZrRu_{1.5}Sb with promising thermoelectric properties. *Adv. Funct. Mater.* **32**, 2200438 (2022).

Supplementary Table 8. Thermoelectric coefficients of state-of-the-art thermoelectric materials and bilayer graphene with circular nanopores.

p -type	n -type	Dimension (mm)	N	ΔT T_c-T_h (K)	Device C_{TE}^* (single strip)	Ref.
Yb _{0.3} Co ₄ Sb ₁₂	Ce _{0.85} Fe ₃ CoSb ₁₂	20×20×14.5	16	574 313-887	10.2% (0.64%)	144
Graphene modified- Yb _y Co ₄ Sb ₁₂	Ce _y Fe ₃ CoSb ₁₂	4×4×12	16	557 300-857	24% (1.5%)	145
Mg _{3.1} Co _{0.1} Sb _{1.5} Bi _{0.49} Te _{0.01}	ZrCoBi _{0.65} Sb _{0.15} Sn _{0.2}	3.6×3.6×4.4	4	400 373-773	10.6% (2.7%)	146
Mg _{0.99} Cu _{0.01} Ag _{0.97} Sb _{0.99}	Mg _{3.2} Sb _{1.5} Bi _{0.49} Te _{0.01} Cu _{0.01}	3×2×2	8	320 279-593	7.3% (0.91%)	147
Ni-doped MgAgSb	Bi ₂ Te ₃	N.A.	2	225 293-568	10.0% (5.0%)	148
Cu ₂ Se	Ni/Ti/Yb/Co ₄ Sb ₁₂	10×4×4	16	680 293-973	9.1% (0.57%)	149
ZrCoBi-based half- Heuslers	N.A.	4.6×1.5×2.4	2	500 323-823	9.0% (4.5%)	150
TaFeSb-based half- Heuslers	N.A.	11×2.6×2.7	2	656 317-973	11.4% (5.7%)	151
(Bi, Sb) ₂ Te ₃	N.A.	N.A.	N.A.	200 298-758	9.0% (0.05%)	152
Yb _{0.09} Ba _{0.05} La _{0.05} Co ₄ Sb ₁₂	Mm _{0.3} Fe _{1.46} Co _{2.54} Sb _{12.05}	4×4×4	64	460 298-758	7.0% (0.11%)	153
N.A.	Reduced graphene oxide	25×5×0.01	6	200 300-500	<0.1% (<0.01%)	This work
N.A.	Graphene prepared by CVD	25×5×0.01	6	200 300-500	<0.1% (<0.01%)	This work
N.A.	2L-GN-Circle	25×5×0.01	6	200 300-500	14.4% (2.4%)	This work

*Thermoelectric coefficient C_{TE} is calculated by $C_{TE} = \frac{T_h - T_c}{T_h} \frac{\sqrt{1 + \overline{ZT}} - 1}{\sqrt{1 + \overline{ZT}} + T_c/T_h}$, where T_h and T_c are the temperature of the hot and cold ends ($\Delta T = T_h - T_c$), respectively. \overline{ZT} is the average ZT over the temperature range, which is calculated by the integration method (i.e., a fourth-order polynomial fitting of the ZT curve).

144. Chu J. *et al.* Electrode interface optimization advances conversion efficiency and stability of thermoelectric device. *Nat. Commun.* **11**, 2723 (2020).

145. Zong, P. A. *et al.* Skutterudite with graphene-modified grain-boundary complexion enhances ZT enabling high-efficiency thermoelectric device. *Energy Environ. Sci.* **10**, 183 (2017).

146. Zhu Q. *et al.* Realizing high conversion efficiency of Mg₃Sb₂-based thermoelectric materials. *J. Power Sources*. **414**, 393 (2019).
147. Liu Z.H. *et al.* Demonstration of ultrahigh thermoelectric efficiency of ~7.3% in Mg₃Sb₂/MgAgSb module for low-temperature energy harvesting. *Joule* **5**, 1196 (2021).
148. Kraemer, D. *et al.* High thermoelectric conversion efficiency of MgAgSb-based material with hot-pressed contacts. *Energy Environ. Sci.* **5**, 1299 (2015).
149. Qiu P. F. *et al.* High-efficiency and stable thermoelectric module based on liquid-like materials. *Joule* **3**, 1538 (2019).
150. Zhu, H.T. *et al.* Discovery of ZrCoBi based half Heuslers with high thermoelectric conversion efficiency. *Nat. Commun.* **9**, 2497 (2018).
151. Zhu, H.T. *et al.* Discovery of TaFeSb-based half-Heuslers with high thermoelectric performance. *Nat. Commun.* **10**, 270 (2019).
152. Pan, Y. *et al.* Melt-centrifuged (Bi,Sb)₂Te₃:Engineering microstructure toward high thermoelectric efficiency. *Adv. Mater.* **30**, 1802016 (2018).
153. Salvador, J. R. *et al.* Conversion efficiency of skutterudite-based thermoelectric modules. *Phys. Chem. Chem. Phys.* **16**, 12510 (2014).”

The improvement of the dielectric loss tangent in the 5G frequency range seems to be the most impacting outcome. An efficient ultra-thin EM shield is certainly of technological value. It seems reasonable to claim that dipole relaxation on hole boundaries is the relevant physical mechanism.

Response: We thank the reviewer for their positive comments regarding the improvement of the dielectric loss tangent of ordered porous graphene in the 5G frequency.

It is to note that several key effects are not properly cited ('dipole relaxation polarization', 'dielectric tangent loss'), which yields a strange feeling of inaccuracy.

Response: We thank the reviewer for their careful reading of our manuscript. We have changed “dipole relaxation polarization” to “dipole polarization relaxation” to describe the dynamic process of dipole moment rearrangements along the external EM field direction. In addition, we have changed “dielectric tangent loss” to “dielectric loss tangent” to refer to the ratio of the imaginary part of the permittivity to the real part ($\tan\delta_e = \epsilon''/\epsilon'$).

To conclude, this work appears more as an engineering project consisting of an interesting optimization of a few-layer graphene film for EM absorption and a more conventional, rather poorly efficient, TE conversion.

Response: We wish to clarify here that the class of ordered nanoporous graphene reported in this work possesses not only a high EM dissipation ability, but also a good thermoelectric performance. For example, the Seebeck coefficient, ZT value, and thermoelectric coefficient of the bilayer graphene with circular nanopores are $-83.0 \mu\text{V/K}$, 0.33, and 14.4%, which are comparable with state-of-the-art thermoelectric materials (e.g., SnTe, Ag₂Se and PbTe in

Supplementary Tables 6 and 8 of the revised Supplementary Information). Furthermore, the overall EM–electricity conversion of the bilayer porous graphene-based device is $\sim 5.6\%$, which is at least three orders of magnitude larger than that of a magnetic FeCo alloy, an outstanding EM absorber but not a good thermoelectric material. Overall, these results lead us to conclude that this class of ordered nanoporous graphene exhibits both high EM dissipation and thermoelectric ability, which may find potential use in EM shielding and absorption, thermoelectrics, and EM–electricity conversion.

The aggregation of numerous studies provided by a large consortium and leading to a plethora of results (more than 100 figures) does not here produce a larger scientific impact.

It finally seems -to fit the format of a paper- that this article should either be reduced to its most exciting part or submitted to a nano / engineering journal.

Response: We agree with the reviewer that such a long Supplementary Information dilutes the key findings in this work. To highlight the most important findings of this work, we have revised Figures 3-5 of the manuscript and significantly shortened the Supplementary Information by reducing its length from 113 to 61 pages and the number of Supplementary Figures from 67 to 34. Accordingly, we refer the reviewer to our revised Supplementary Information.

Reviewer 3:

The authors proposed an energy harvesting material from electromagnetic waves to electricity by using Graphene. The graphene is heated by induction, then electricity is generated by thermoelectric effects. The porous graphene shows a higher performance than the normal graphene due to enhanced thermoelectric properties. It is interesting.

Response: We thank the reviewer for their positive comments regarding the novelty of our work.

However, the generated power is 1.5 microW under the 1000W electromagnetic wave irradiation. If the proposed graphene is used for energy harvesting, the irradiated power is in the order of 10mW at most. Only 1nW generated power might be too low for practical use even if it is for energy harvesting.

Response: We wish to clarify that, in our original manuscript, the power of our EM source is 100 W, not 1,000 W (page 21, line 3 of original Supplementary Information). To address the reviewer’s comment regarding the EM–electricity conversion of our ordered porous graphene-based device, we have estimated the overall EM–electricity conversion using the following equation:

$$x = \frac{W_{\text{out}}}{\frac{V_{\text{device}}}{V_{\text{chamber}}} W_{\text{source}} \alpha}$$

where W_{out} is the output work of the graphene device (0.23 mJ during 180 s of EM radiation), W_{source} is the output work of the EM generator, α is the attenuation degree, and V_{device} and V_{chamber} represent the volume of the graphene device and EM radiation chamber, respectively. Note that it would be inaccurate to use the total electric power output to calculate the overall EM–electricity conversion since only a small fraction of the EM waves is absorbed by the graphene-based device. Due to the large EM wavelength, the EM wave attenuation is negligible within the chamber ($\alpha \approx 1$), leading to an almost constant EM intensity in the chamber. Therefore, the fraction of the EM waves exposed to the graphene device can be calculated by the volume fraction of the graphene device relative to the chamber volume. Our graphene device's overall EM–electricity conversion was calculated to be $\sim 5.6\%$. In, addition, the volumetric power density was measured to be 204 W/m^3 , which is about two orders of magnitude higher than current inorganic TE materials (e.g., $\text{Ag}_2\text{Se/PEDT}$, $\text{NbSe}_2/\text{WSe}_2$, CuSe_2 etc.; See Supplementary Table 7 of the revised Supplementary Information). These results suggest that the ordered porous graphene-based device has the potential for efficient EM–electricity conversion. To clarify this point, we have added the following text to the revised manuscript:

“The overall EM–electricity conversion of porous bilayer graphene is estimated to be $\sim 5.6\%$ (Supplementary Note 10).”

We have revised Figure 5d of the revised manuscript:

“(d) Power density (P_{density}) as a function of time.”

We have added Supplementary Note 10 to the revised Supplementary Information:

“**Supplementary Note 10. Calculation of the overall EM–electricity conversion of ordered nanoporous graphene-based devices**

The overall EM–electricity conversion (x) of our ordered porous graphene-based device can be calculated as:

$$x = \frac{W_{\text{out}}}{\frac{V_{\text{device}}}{V_{\text{chamber}}} W_{\text{source}} \alpha} \quad (13)$$

where W_{out} is the output work of the graphene device (0.23 mJ during 180 s of EM radiation), W_{source} is the output work of the EM generator (100 W), α is the attenuation degree, and V_{device} and V_{chamber} represent the volume of the graphene device (six graphene strips; each strip is 25 mm \times 5 mm \times 0.01 mm) and EM radiation chamber (447 mm \times 247 mm \times 281 mm), respectively. It is well established that the GHz-frequency EM wave attenuation is negligible in a chamber with these dimensions ($\alpha \approx 1$), which gives rise to an almost constant EM intensity in the chamber^{31,32}. Therefore, the fraction of the EM waves exposed to the graphene device can be calculated by the volume fraction of the graphene device relative to the chamber volume. As a result, the overall EM–electricity conversion of the bilayer porous graphene-based device was calculated to be 5.6 %.”

We have added volumetric P_{density} to Supplementary Table & of the revised Supplementary

Information:

“**Supplementary Table 7.** Maximum power density and temperature difference of current electricity generating materials that utilize temperature gradients.

Material	P_{density} (W/m ³)	ΔT (K)	$P_{\text{density}} / \Delta T$ (W/m ³ ·K)	Ref.
n/p -SWCNT	4.16	30	0.138	124
TENG	~ 1.4	80	1.75×10^{-2}	125
n -type Ag ₂ Se	23	30	7.66×10^{-1}	126
PEDOT:PSS/Cu ₂ Se	91	30	3.0	127
TE/PEDOT:PSS	572	40	14.3	128
PANI/CNT/Te	62.4	40	1.56	129
PEDOT:PSS/TiS ₂ [(HA)(NMF)]	80.0	40	2.0	130
PEDOT:PSS/Te/Cu ₇ Te ₄	39.5	39.1	1.01	131
PVP/Ag/Ag ₂ Te	34.1	39.6	8.6×10^{-1}	132
Ag ₂ Se/Ag/PEDOT	74.7	27	2.76	133
CNT/PEG	12.5	25	0.5	134
NbSe ₂ /WS ₂	7.3×10^{-1}	60	1.21×10^{-3}	135
Bi _{0.5} Sb _{0.15} Te ₃	22.8	N.A.	N.A.	136
(Nb _{0.8} Ta _{0.2}) _{0.8} Ti _{0.2} FeSb	211	655	0.32	137
Ag-Se-TENG	32.1	110	0.29	138
Bi ₂ Te ₃ -TENG	92.0	52.5	1.75	139
Ce ₇ Fe ₃ CoSb ₁₂	21.0	570	3×10^{-2}	140
Bi ₂ Te ₃	11.4	N.A.	N.A.	141
Ge _{0.84} Pb _{0.1} Sb _{0.06}	176.0	425	0.41	142
Se-Co-Ni alloy	43.6	500	0.87	143

”

In addition, we have added the following references to the revised Supplementary Information:

- “31. Toyoshima, M. *et al.* Comparison of microwave and light wave communication systems in space applications. *Opt. Eng.* 47, 015003 (2007).
 32. Heremans J. P. *et al.* Enhancement of thermoelectric efficiency in PbTe by distortion of the electronic density of states. *Science* 321, 554 (2008).”

Furthermore, we comment here that besides the EM–electricity conversion, this class of

ordered nanoporous graphene exhibits an outstanding EM dissipation (see Supplementary Table 2 of the revised manuscript) and thermoelectric performance. We have calculated the ZT value of the ordered nanoporous graphene using $ZT = \sigma S^2 T / \kappa_T$ (where S , σ , T , and κ_T are the Seebeck coefficient, electric conductivity, absolute temperature, and thermal conductivity, respectively) and the thermoelectric coefficient (C_{TE}) of ordered porous graphene as:

$$C_{TE} = \frac{T_h - T_c}{T_h} \frac{\sqrt{1 + ZT} - 1}{\sqrt{1 + ZT} + T_c / T_h}$$

where T_h and T_c are the temperature of the hot and cold ends ($\Delta T = T_h - T_c$), respectively. In comparison to other materials, we note that both the ZT value and the thermoelectric coefficient of the bilayer graphene with ordered circular nanopores are at least three orders of magnitude larger than that of pristine and defective graphene and comparable with state-of-the-art thermoelectric materials. These results lead us to suggest that this class of ordered nanoporous graphene can find potential use in a wide range of applications in addition to EM–electricity conversion, for example, EM absorption and shielding, and thermoelectrics through the EM-induced overheating of devices. To clarify this point, we have added the following text to the revised manuscript:

“We further evaluated the thermoelectric conversion ability of ordered porous graphene by calculating the ZT value ($ZT = \sigma S^2 T / \kappa_T$)³⁷ and the thermoelectric coefficient³⁷. As shown in **Fig. 4b** and **Supplementary Fig. 26**, we calculated the largest dimensionless figure of merit ZT ($ZT = S^2 \sigma T / \kappa_T$) to be 0.33 over a temperature range of 300–500 K. We note that both the ZT value and the thermoelectric coefficient of the bilayer graphene with ordered circular nanopores are about three orders of magnitude higher than that of bilayer nonporous graphene, significantly higher than other graphene-based composites, and comparable to current state-of-the-art thermoelectric materials (**Supplementary Tables 6 and 8**).”

“This novel class of multifunctional ordered nanoporous graphene may expand the potential utility of graphene in energy harvesting, thermal management, EM shielding and absorption, thermoelectrics, and photocatalytic water splitting.”

We have added Figure 4b to the revised manuscript:

Fig. 4. Effect of staggered, ordered nanopores on the Seebeck effect of graphene. (a) Temperature-dependent Seebeck coefficient (S) and **(b) ZT value** of bilayer graphene with and without circular nanopores. Error bars represent standard deviations from three independent measurements.”

We have added Supplementary Figure 26, Supplementary Tables 6 and 8, and the following references to the revised Supplementary Information:

Supplementary Fig. 26 - Temperature-dependent Seebeck coefficient (S) and ZT value of graphene with different numbers of layers and pore shapes. (a-d) S and (e-h) ZT value of porous graphene with (a, e) 1, (b, f) 2, (c, g) 3, and (d, h) 6 layers as a function of temperature.

Supplementary Table 6. ZT values of conventional thermoelectric materials and graphene.

Sample	Maximum ZT (Temperature range)	ZT at 500K	Ref.
Single layer graphene	<0.01 at 600K (300K-600K)	<0.01	108
Defective graphene	<0.01 at 300K (300K)	N.A.	109
Graphene/ Bi_2Te_3	0.55 at 500K (300K-500K)	0.55	110
Graphene/ CoSb_3	0.61 at 800K (300K-800K)	~0.15	111
Graphene/ $\text{Bi}_{1-x}\text{Te}_3$	0.2 at 480K (300K-480K)	N.A.	112
Graphene/ CuInTe_2	0.4 at 700K (300K-700K)	0.1	113
Cu_2Se	1.2 at 1000K (300K-1000K)	~0.1	114
AgMnSbTe_3	1.4 at 800K (300K-800K)	~0.6	115
Ga-doped PbTe	1.55 at 723K (300K-723K)	~0.5	116
AgMnGeSbTe_4	1.05 at 773K (300K-773K)	~0.6	117
$\text{Pb}_7\text{Bi}_4\text{Se}_{13}$	1.35 at 800K (300K-800K)	~0.5	118
(Pb, Ge, Sb, Cd)-doped SnTe	1.5 at 800K (300K-800K)	~0.6	119

Cu ₁₂ Sb ₄ S ₁₃ -based alloy	1.15 at 723K (300K-723K)	0.4	120
Cu ₂ SnSe ₃	1.6 at 823K (300K-823K)	<0.2	121
Bi ₂ Si ₂ Te ₆	0.51 at 623K (300K-623K)	<0.3	122
ZrRu _{1+x} Sb	0.2 at 973K (300K-973K)	<0.1	123
2L-GN-Circle	0.33 at 500K (300K-500K)	0.33	This work

108. Reshak, A. *et al.* Thermoelectric properties of a single graphene sheet and its derivatives. *J. Mater. Chem. C* **2**, 2356 (2014).
109. Anno, Y. *et al.* Enhancement of graphene thermoelectric performance through defect engineering. *2D Mater.* **4**, 025019 (2017).
110. Ahmad, K. *et al.* Enhanced thermoelectric performance of Bi₂Te₃ based graphene nanocomposites. *Appl. Surf. Sci.* **2**, 474 (2019).
111. Feng, B. *et al.* Enhanced thermoelectric properties of *p*-type CoSb₃/graphene nanocomposite. *J. Mater. Chem. A* **1**, 13111 (2013).
112. Liang, B. B. *et al.* Fabrication and thermoelectric properties of graphene/Bi₂Te₃ composite materials. *J. Nanomater.* **6**, 6 (2013).
113. Chen, H.J. *et al.* Thermoelectric properties of CuInTe₂/graphene composites. *CrystEngComm* **15**, 6648 (2013).
114. Choo, S. *et al.* Cu₂Se-based thermoelectric cellular architectures for efficient and durable power generation. *Nat. Commun.* **12**, 3550 (2021).
115. Luo, Y.B. *et al.* Cubic AgMnSbTe₃ semiconductor with a high thermoelectric performance. *J. Am. Chem. Soc.* **143**, 13990 (2021).
116. Luo, Z.Z. *et al.* Extraordinary role of Zn in enhancing thermoelectric performance of G-doped *n*-type PbTe. *Energy Environ. Sci.* **15**, 368 (2022).
117. Ma, Z. *et al.* High entropy semiconductor AgMnGeSbTe₄ with desirable thermoelectric performance. *Adv. Funct. Mater.* **21**, 2103197 (2013).
118. Hu, L. *et al.* High thermoelectric performance enabled by convergence of nested conduction bands in Pb₇Bi₄Se₁₃ with low thermal conductivity. *Nat. Commun.* **12**, 4793 (2021).
119. Zhang, Q. *et al.* High-performance thermoelectric material and module driven by medium-entropy engineering in SnTe. *Adv. Funct. Mater.* **32**, 220548 (2022).
120. Hu, H.H. *et al.* Thermoelectric Cu₁₂Sb₄S₁₃-based synthetic minerals with a sublimation-derived porous network. *Adv. Mater.* **33**, 2103633 (2021).
121. Hu, L. *et al.* High thermoelectric performance through crystal symmetry enhancement in triply doped diamondoid compound Cu₂SnSe₃. *Adv. Energy Mater.* **11**, 2100661 (2021).
122. Luo, Y. B. *et al.* Thermoelectric performance of 2D Bi₂Si₂Te₆ Semiconductor. *J. Am. Chem. Soc.* **144**, 1445 (2022).
123. Wang, L.Y. *et al.* Discovery of a slater-pauling semiconductor ZrRu_{1.5}Sb with promising thermoelectric properties. *Adv. Funct. Mater.* **32**, 2200438 (2022).

Supplementary Table 8. Thermoelectric coefficients of state-of-the-art thermoelectric materials and bilayer graphene with circular nanopores.

p -type	n -type	Dimension (mm)	N	ΔT T_c-T_h (K)	Device C_{TE}^* (single strip)	Ref.
Yb _{0.3} Co ₄ Sb ₁₂	Ce _{0.85} Fe ₃ CoSb ₁₂	20×20×14.5	16	574 313-887	10.2% (0.64%)	144
Graphene modified- Yb _y Co ₄ Sb ₁₂	Ce _y Fe ₃ CoSb ₁₂	4×4×12	16	557 300-857	24% (1.5%)	145
Mg _{3.1} Co _{0.1} Sb _{1.5} Bi _{0.49} Te _{0.01}	ZrCoBi _{0.65} Sb _{0.15} Sn _{0.2}	3.6×3.6×4.4	4	400 373-773	10.6% (2.7%)	146
Mg _{0.99} Cu _{0.01} Ag _{0.97} Sb _{0.99}	Mg _{3.2} Sb _{1.5} Bi _{0.49} Te _{0.01} Cu _{0.01}	3×2×2	8	320 279-593	7.3% (0.91%)	147
Ni-doped MgAgSb	Bi ₂ Te ₃	N.A.	2	225 293-568	10.0% (5.0%)	148
Cu ₂ Se	Ni/Ti/Yb/Co ₄ Sb ₁₂	10×4×4	16	680 293-973	9.1% (0.57%)	149
ZrCoBi-based half- Heuslers	N.A.	4.6×1.5×2.4	2	500 323-823	9.0% (4.5%)	150
TaFeSb-based half- Heuslers	N.A.	11×2.6×2.7	2	656 317-973	11.4% (5.7%)	151
(Bi, Sb) ₂ Te ₃	N.A.	N.A.	N.A.	200 298-758	9.0% (0.05%)	152
Yb _{0.09} Ba _{0.05} La _{0.05} Co ₄ Sb ₁₂	Mm _{0.3} Fe _{1.46} Co _{2.54} Sb _{12.05}	4×4×4	64	460 298-758	7.0% (0.11%)	153
N.A.	Reduced graphene oxide	25×5×0.01	6	200 300-500	<0.1% (<0.01%)	This work
N.A.	Graphene prepared by CVD	25×5×0.01	6	200 300-500	<0.1% (<0.01%)	This Work
N.A.	2L-GN-Circle	25×5×0.01	6	200 300-500	14.4% (2.4%)	This work

*Thermoelectric coefficient C_{TE} is calculated by $C_{TE} = \frac{T_h - T_c}{T_h} \frac{\sqrt{1 + \overline{ZT}} - 1}{\sqrt{1 + \overline{ZT}} + T_c/T_h}$, where T_h and T_c are the temperature of the hot and cold ends ($\Delta T = T_h - T_c$), respectively. \overline{ZT} is the average ZT over the temperature range, which is calculated by the integration method (i.e., a fourth-order polynomial fitting of the ZT curve).

144. Chu J. *et al.* Electrode interface optimization advances conversion efficiency and stability of thermoelectric device. *Nat. Commun.* **11**, 2723 (2020).

145. Zong, P. A. *et al.* Skutterudite with graphene-modified grain-boundary complexion enhances ZT enabling high-efficiency thermoelectric device. *Energy Environ. Sci.* **10**, 183 (2017).

146. Zhu Q. *et al.* Realizing high conversion efficiency of Mg₃Sb₂-based thermoelectric materials. *J. Power Sources*. **414**, 393 (2019).
147. Liu Z.H. *et al.* Demonstration of ultrahigh thermoelectric efficiency of ~7.3% in Mg₃Sb₂/MgAgSb module for low-temperature energy harvesting. *Joule* **5**, 1196 (2021).
148. Kraemer, D. *et al.* High thermoelectric conversion efficiency of MgAgSb-based material with hot-pressed contacts. *Energy Environ. Sci.* **5**, 1299 (2015).
149. Qiu P. F. *et al.* High-efficiency and stable thermoelectric module based on liquid-like materials. *Joule* **3**, 1538 (2019).
150. Zhu, H.T. *et al.* Discovery of ZrCoBi based half Heuslers with high thermoelectric conversion efficiency. *Nat. Commun.* **9**, 2497 (2018).
151. Zhu, H.T. *et al.* Discovery of TaFeSb-based half-Heuslers with high thermoelectric performance. *Nat. Commun.* **10**, 270 (2019).
152. Pan, Y. *et al.* Melt-centrifuged (Bi,Sb)₂Te₃:Engineering microstructure toward high thermoelectric efficiency. *Adv. Mater.* **30**, 1802016 (2018).
153. Salvador, J. R. *et al.* Conversion efficiency of skutterudite-based thermoelectric modules. *Phys. Chem. Chem. Phys.* **16**, 12510 (2014).”

REVIEWER COMMENTS

Reviewer #1 (Remarks to the Author):

None

Reviewer #2 (Remarks to the Author):

An extensive answer was provided to justify the high TE efficiency with a ZT of 0.33 and a coefficient of performance of 14%. The computed ZT values lie in the value range of the state-of-the-art. The temperatures T_c and T_h used for the coefficient of performance should be clearly indicated.

This quite considerable TE performance is somehow surprising when considering the small output power of 1microW.

The low incoming EM power on the film might explain this outcome, but this argument does not really lift worries about the final relevance of such a film for EMI harvesting.

Overall, the paper might be considered for publication if the reasons for the low final output electrical power are clearly elucidated.

Reviewer #3 (Remarks to the Author):

Thank you for your response. I am sorry that I completely misunderstood the irradiated power on the graphene film. I have no questions about the manuscript. Again, the present results are interesting.

Reviewer 2:

An extensive answer was provided to justify the high TE efficiency with a ZT of 0.33 and a coefficient of performance of 14%. The computed ZT values lie in the value range of the state-of-the-art. The temperatures T_c and T_h used for the coefficient of performance should be clearly indicated.

Response: We are glad that the reviewer agrees with us that the calculated ZT value of our porous graphene lies in the range of state-of-the-art thermoelectric materials. We wish to clarify that in our last revision, we added Supplementary Table 8 to clearly indicate the T_h and T_c used in calculating the thermoelectric coefficient of devices based on porous graphene and state-of-the-art thermoelectric materials. For instance, for bilayer graphene with circular nanopores (last row of Supplementary Table 8), T_c-T_h (K) is 300–500. Accordingly, we refer the reviewer to our revised Supplementary Information.

This quite considerable TE performance is somehow surprising when considering the small output power of 1 microW. The low incoming EM power on the film might explain this outcome, but this argument does not really lift worries about the final relevance of such a film for EMI harvesting. Overall, the paper might be considered for publication if the reasons for the low final output electrical power are clearly elucidated.

Response: The overall EM–electricity conversion was calculated to be $\sim 5.6\%$ and the volumetric power density was measured to be 204 W/m^3 , which is about two orders of magnitude higher than current inorganic TE materials (e.g., $\text{Ag}_2\text{Se/PEDT}$, $\text{NbSe}_2/\text{WSe}_2$, CuSe_2 , etc.; see Supplementary Table 7 of the last revision of Supplementary Information). We wish to clarify that the small output power of our device ($\sim 1.5 \mu\text{W}$) is attributed to both low incoming EM power (as mentioned by the reviewer) and the small dimension of our device. We hypothesize that the output power depends on the device dimension at a given incoming EM power. To address the reviewer’s comment, we have performed additional experiments to study the effect of the thickness and number of graphene strips on the device’s output power. We found that by increasing the strip thickness (from $10 \mu\text{m}$ to $25 \mu\text{m}$) and number (from 6 to 30), the device’s output power reached $10.5 \mu\text{W}$, which is a one order of magnitude increase. This result shows that the device dimension can be easily scaled up for specific practical applications.

To clarify this point, we have added the following text to the revised manuscript:

“We comment here that the device’s output power can be enhanced by increasing the thickness and number of the porous graphene strips (**Supplementary Note 10** and **Supplementary Fig. 31**).”

In addition, we have added the following text and Supplementary Fig. 31 to the revised Supplementary Information:

“Supplementary Note 10. Effect of thickness and number of ordered nanoporous graphene strips on device output power

With a constant incoming EM power, we hypothesize that the ordered nanoporous graphene-based device’s output power depends on (i) the thickness and (ii) the number of the nanoporous graphene strip.

i) Effect of the thickness of the strips on the device’s output power

As shown in **Supplementary Fig. 31a**, as the thickness of the ordered nanoporous graphene strip increases, the device output power first increases and then decreases and the maximum output power reaches $\sim 3.1 \mu\text{W}$, almost three times that of a $10 \mu\text{m}$ -thick ordered nanoporous graphene strip. According to the EM shielding mechanism, with an increase in the EM absorbing material thickness, the EM wave dissipation path increases and the inverse radiation decreases, which is consistent with our experimental observation that the EM absorption increases monotonically from 11% to 29% as the thickness increases from $10 \mu\text{m}$ to $50 \mu\text{m}$ (**Supplementary Fig. 31a**). The decrease in output power with an increase in thickness from $25 \mu\text{m}$ to $50 \mu\text{m}$ is attributed to the increase in the internal resistance of the device.

ii) Effect of the number of strips on the device’s output power

As illustrated in **Supplementary Fig. 31b**, the output power of the device increases from $1.5 \mu\text{W}$ to $10.5 \mu\text{W}$ as the number of ordered nanoporous graphene strips increases from 6 to 30. This increase in output power is due to the increase in device volume as the number of ordered nanoporous graphene strips increases, leading to an increase in the amount of absorbed EM waves. These results suggest that the device output power can be increased by increasing the surface area and the number of ordered nanoporous graphene strips.

Supplementary Fig. 31 – Effect of device dimension on the output power. (a) EM absorption and output power (P_{out}) of the ordered nanoporous graphene-based device as a function of the thickness of the ordered nanoporous graphene strips. The device consisted of six ordered nanoporous graphene strips with each strip being 25 mm long and 5 mm wide. The thickness

was varied from 10 to 25 μm . (b) P_{out} of the ordered nanoporous graphene-based device as a function of the number of strips. Each strip was 25 mm long, 5 mm wide, and 25 μm thick.”

REVIEWERS' COMMENTS

Reviewer #2 (Remarks to the Author):

The rebuttal is satisfactory and the paper deserves publication in Nature Communication.